# ThinK: Thinner Key Cache by Query-Driven Pruning

Yuhui Xu[1]    Zhanming Jie[1]    Hanze Dong[1]    Lei Wang[1]    Xudong Lu[2]
Aojun Zhou[2]    Amrita Saha[1]    Caiming Xiong[1]    Doyen Sahoo[1]

[1]Salesforce AI Research    [2] The Chinese University of Hong Kong

## ABSTRACT

Large Language Models (LLMs) have revolutionized the field of natural language processing, achieving unprecedented performance across a variety of applications. However, their increased computational and memory demands present significant challenges, especially when handling long sequences. This paper focuses on the long-context scenario, addressing the inefficiencies in KV cache memory consumption during inference. Unlike existing approaches that optimize the memory based on the sequence length, we identify substantial redundancy in the channel dimension of the KV cache, as indicated by an uneven magnitude distribution and a low-rank structure in the attention weights. In response, we propose ThinK, a novel query-dependent KV cache pruning method designed to minimize attention weight loss while selectively pruning the least significant channels. Our approach not only maintains or enhances model accuracy but also achieves a reduction in KV cache memory costs by over 20% compared with vanilla KV cache eviction and quantization methods. For instance, ThinK integrated with KIVI can achieve $2.8\times$ peak memory reduction while maintaining nearly the same quality, enabling a batch size increase from $4\times$ (with KIVI alone) to $5\times$ when using a single GPU. Extensive evaluations on the LLaMA and Mistral models across various long-sequence datasets verified the efficiency of ThinK. Our code has been made available at `https://github.com/SalesforceAIResearch/ThinK`.

## 1 INTRODUCTION

Large language models (LLMs) (Hadi et al., 2023; Brown et al., 2020; OpenAI, 2023; Touvron et al., 2023a;b; Scao et al., 2022; Reid et al., 2024) have emerged as a dominant paradigm in natural language processing, achieving state-of-the-art performance across various tasks. A key principle, the Scaling Law (Kaplan et al., 2020), suggests that LLMs exhibit emergent abilities as model size increases, improving their capacity to understand complex context and handle long sequences (Xiong et al., 2023). This growth in capacity enables LLMs to generate coherent, contextually accurate responses and supports a variety of downstream applications, such as document summarization (Zhang et al., 2019; 2024a), code generation (Chen et al., 2021b), solving mathematical problems (Hendrycks et al., 2021; Zhou et al., 2023; Wang et al., 2023; Lightman et al., 2023), and conversational AI (OpenAI, 2022; 2023).

Despite their success in various applications, generating outputs with LLMs incurs significant computational and financial costs, which rise with increasing model size and sequence length. Both the training (Strubell et al., 2020; Hoffmann et al., 2022; Dong et al., 2024a) and inference (Ainslie et al., 2023) stages involve frequent generation, further contributing to these costs. Consequently, efficient LLMs have gained traction in recent years (Hu et al., 2021; Wan et al., 2023). To address these challenges, quantization (Frantar et al., 2022; Lin et al., 2024; Dettmers et al., 2024; Xu et al., 2023) and pruning methods (Sun et al., 2023; Frantar & Alistarh, 2023; Lu et al., 2024b) are employed to reduce model size. Additionally, the key-value (KV) cache, stored in GPU memory alongside model parameters, scales linearly with both sequence length and batch size, creating a substantial memory burden when handling long sequences. Consequently, effective management of extended contexts is essential for the practical deployment of LLMs. In this paper, we focus on the long-context scenario, aiming to reduce memory consumption associated with processing lengthy sequences.

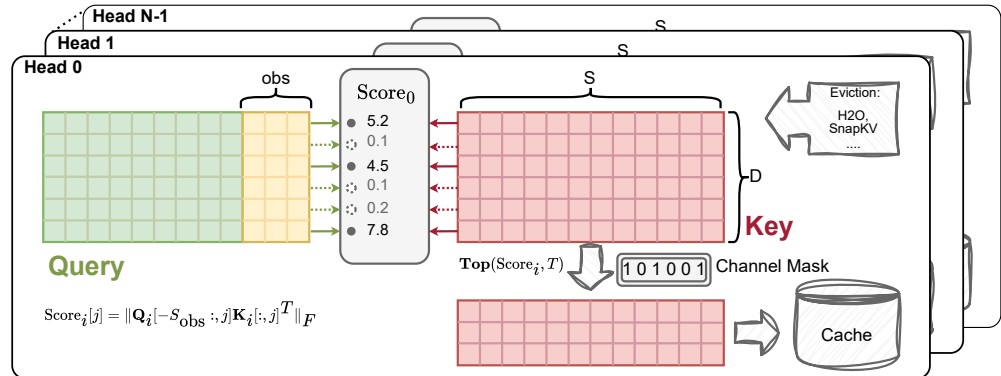

Figure 1: An illustration of the pruning procedure of THINK. Within each attention head, scores are computed for each channel, and only the top $T$ channels out of $D$ are selected for retention. A binary channel mask, along with the pruned keys, is then stored in the cache memory.

Specifically, the number of KV cache parameters is the product of batch size $B$, sequence length $S$, number of layers $L$, number of heads $N$, channel size per head $D$, *i.e.*, $\mathbf{K}, \mathbf{V} \in \mathbb{R}^{B \times S \times L \times N \times D}$, which need to be stored in the GPU memory during inference. To reduce memory and computational costs during inference, efficiency can only be achieved by pruning the dimensions across $S, L, N, D$ or applying quantization to the caches. It is well-acknowledged that token importance tends to be sparse. Consequently, KV eviction algorithms have been proposed to reduce the memory footprint by addressing the sequence length dimension $S$ (Xiao et al., 2023b; Li et al., 2024; Zhang et al., 2024c; Leviathan et al., 2023). Additionally, inter-layer redundancy has been explored (Liu et al., 2024a; Wu & Tu, 2024; Brandon et al., 2024) to address the layer dimension $L$. Despite these advances, existing methods have largely overlooked the channel dimension $D$. In this paper, we highlight that the magnitudes across key cache channel dimensions are significantly imbalanced, and we observe a low-rank structure in attention weights. Based on these findings, we hypothesize that the channel dimension of the key cache exhibits redundancy. Consequently, we focus on exploring the redundancy in the KV cache along dimension $D$, aiming to develop strategies that reduce memory costs without compromising performance.

In this paper, we introduce THINK, a simple yet effective method for KV cache pruning. To pinpoint the least significant channels, we formulate the problem as an optimization task, aiming to minimize the loss in attention weights caused by pruning. To effectively address this problem, we propose a novel query-dependent criterion that assesses the importance of each channel. Using this criterion, we then select the most critical channels in a greedy fashion. We evaluate THINK using the LLaMA (Meta, 2024) and Mistral (Jiang et al., 2023) models, and validate its effectiveness across various long-sequence datasets. The results indicate that, when paired with token eviction and KV cache quantization methods, THINK not only maintains comparable or superior accuracy but also reduces KV cache memory costs by over 20%.

**Contributions.** This work pioneers the investigation into the sparsity within the channels of the KV cache. Specifically, we uncover that the activated key cache is sparse for a given query. This insight allows us to prune the key cache channels using a query-induced norm. Building on this insight, we introduce THINK, the first channel pruning method specifically designed for KV cache. THINK reduces the dimensionality of the cache channels, leading to linear savings in memory usage. As a plug-and-play technique, THINK is orthogonal to other KV cache compression schemes (*e.g.* KV cache eviction, quantization). Our extensive experiments demonstrate THINK's remarkable efficiency on the LLaMA and Mistral models. Moreover, we explore the potential extension of THINK to value cache pruning (THINKV), showcasing the broad applicability of our method.

## 2 OBSERVATIONS

We identify several key observations that motivate our approach to pruning the channels of the KV cache. Specifically, we visualize the magnitude of the KV cache and perform singular value decomposition (SVD) on the attention mechanism of the LLaMA model.

**Magnitudes of KV cache channels.** Figure 4 (in Appendix A) visualizes the absolute values of the KV cache across tokens in each channel[1]. Consistent with previous findings (Lin et al., 2024; Xiao et al., 2023a; Liu et al., 2024b), we observe that only certain channels have significant magnitudes in the key cache, whereas the value cache lacks obvious patterns. For instance, in layer 14 (Figure 4 (a)), the magnitudes in the key cache are substantially higher around the $50^{th}$ channel across all tokens. A similar pattern is observed in the $50^{th}$ and $150^{th}$ channels of the first head in layer 20 (Figure 4 (c)). Given such an observation, Liu et al. (2024b) proposed to perform quantization over the channels of the key cache. Beyond quantization, our findings suggest that certain key cache channels with smaller contributions to the attention mechanism can be pruned. Moreover, channel quantization and pruning are orthogonal techniques, meaning they can be applied concurrently to further improve model efficiency.

**Singular value analysis.** We conducted singular value decomposition (SVD) (Demmel, 1997) on the attention weights of the specified layers to investigate their principal components. The singular values derived from SVD capture the effective rank of the attention matrix, indicating how the information is distributed across different components.

Figure 5 (a) (in Appendix A) illustrates the energy distribution of the singular values, plotted on a logarithmic scale to enhance visibility of the differences. Notably, only a few singular values exhibit high energy levels exceeding 0.01 across all heads and layers, highlighting their relative significance. This observation aligns with previous findings (Bhojanapalli et al., 2021), where a small subset of singular values often captures most of the information in attention mechanisms. In addition, the rapid decay of the energy suggests that a low-rank approximation can effectively capture the essential information in the key cache.

Figure 5 (b) (in Appendix A), the normalized cumulative energy sum reveals that the top 50 singular values account for over 90% of the total energy. These findings suggest that the attention matrix is inherently low-rank (Wang et al., 2020; Chen et al., 2021a; Dong et al., 2024b), indicating that the key cache can be approximated using low-dimensional vectors (Singhania et al., 2024).

## 3 THINK

**Notations.** We use uppercase letters (*e.g.*, $X, Y$) to denote scalar values and boldface uppercase letters (*e.g.*, $\mathbf{Q}, \mathbf{K}$) to denote matrices and tensors. The notation $\|\cdot\|_p$ denotes the $l_p$-norm for vectors. Unless otherwise specified, $\|\cdot\|$ denotes the $l_2$-norm. The Frobenius norm is denoted by $\|\cdot\|_F$. The floor function is denoted by $\lfloor\cdot\rfloor$, and the ceiling function is denoted by $\lceil\cdot\rceil$.

### 3.1 PRELIMINARY STUDY OF KV CACHE OPTIMIZATION

In scenarios with extended contexts or batch processing, the main limitations in terms of memory and speed are due to the handling of the KV cache size. Considering a batch of requests to a Large Language Model (LLM) service that provides a long input prompt consisting of tokens $[x_{B1}, ..., x_{BS}]$, the total KV cache size can be computed as follows: $2 \times B \times S \times L \times N \times D$, where $L$ is the number of layers, $N$ is the number of heads, $D$ is the head dimension. The KV cache size grows linearly as the batch size $B$ and sequence length $S$. For a model with multihead attention (MHA) (Vaswani et al., 2017), such as LLaMA2-7B (Touvron et al., 2023b), a context length of 2048 and a batch size of 13 require storing a 13 GB KV cache, which is equivalent to the size of the model parameters. The KV cache must be transferred from off-chip memory (HBM) (Jia et al., 2018) to on-chip memory (cache) for each token generated, leading to a memory bottleneck. This substantial memory demand highlights the challenges in managing large-scale models and the need for efficient memory utilization strategies. Current methods optimize the KV cache based on the sequence length $S$ (Xiao et al., 2023b; Zhang et al., 2024c; Li et al., 2024) and precision (Hooper et al., 2024; Liu et al., 2024b). We will introduce a new method, THINK, to optimize it from the perspective of the number of head dimensions $D$.

**Magnitude based Pruning:** Based on the observations in Figure 4 which depicts the significant variation in the magnitudes across different channels, one straightforward criterion is to use the

---

[1]We use the visualization code from `https://github.com/jy-yuan/KIVI/tree/main/vis`.

Table 1: Performance comparison of pruning key cache by $l_p$ norm on LongBench.

| Method | λ | Single-Document QA | | | Multi-Document QA | | | Summarization | | | Few-shot Learning | | | Synthetic | | Code | | Avg. |
|---|---|---|---|---|---|---|---|---|---|---|---|---|---|---|---|---|---|---|
| | | NrtvQA | Qasper | MF-en | HotpotQA | 2WikiMQA | Musique | GovReport | QMSum | MultiNews | TREC | TriviaQA | SAMSum | PCount | PRe | Lcc | RB-P | |
| H2O | 0.0 | 23.52 | 17.93 | 34.68 | 42.11 | 33.52 | 19.92 | 22.11 | 22.56 | 23.82 | 41.00 | 90.46 | 40.20 | 5.87 | 69.50 | 56.71 | 51.69 | 37.23 |
| +$l_1$ | 0.3 | 23.38 | 17.15 | 34.99 | 40.56 | 31.49 | 19.90 | 21.37 | 22.13 | 23.44 | 40.50 | 90.10 | 40.65 | 5.41 | 69.00 | 58.64 | 54.99 | 37.11 |
| +$l_1$ | 0.4 | 23.51 | 15.40 | 34.37 | 40.71 | 31.28 | 20.24 | 21.25 | 22.29 | 22.54 | 38.50 | 89.22 | 39.27 | 5.87 | 68.33 | 58.47 | 54.33 | 36.60 |
| +$l_2$ | 0.3 | 23.98 | 17.04 | 35.19 | 39.27 | 31.29 | 20.40 | 21.62 | 22.46 | 23.34 | 40.50 | 89.75 | 40.71 | 5.54 | 68.67 | 60.12 | 58.52 | 37.40 |
| +$l_2$ | 0.4 | 23.76 | 16.23 | 32.19 | 40.23 | 32.13 | 20.69 | 21.30 | 22.25 | 23.20 | 39.50 | 89.61 | 40.24 | 5.66 | 69.00 | 60.09 | 59.45 | 37.22 |
| SnapKV | 0.0 | 24.84 | 23.96 | 38.77 | 42.75 | 34.55 | 20.87 | 22.26 | 22.61 | 23.97 | 70.00 | 90.52 | 40.29 | 5.81 | 69.50 | 59.04 | 51.81 | 40.10 |
| +$l_1$ | 0.3 | 24.43 | 24.63 | 40.11 | 41.83 | 33.47 | 21.22 | 21.47 | 22.41 | 23.73 | 66.50 | 90.39 | 40.20 | 5.70 | 68.10 | 61.04 | 55.37 | 40.04 |
| +$l_1$ | 0.4 | 24.58 | 24.87 | 39.30 | 42.76 | 31.95 | 20.47 | 20.95 | 22.22 | 23.42 | 55.50 | 90.22 | 39.13 | 5.82 | 68.39 | 60.71 | 56.10 | 39.15 |
| +$l_2$ | 0.3 | 24.47 | 24.73 | 38.16 | 41.86 | 32.23 | 20.23 | 21.59 | 22.45 | 23.77 | 67.50 | 90.33 | 40.31 | 5.70 | 68.42 | 62.65 | 60.07 | 40.28 |
| +$l_2$ | 0.4 | 24.52 | 23.75 | 38.35 | 42.42 | 32.96 | 20.39 | 21.21 | 22.28 | 23.41 | 60.00 | 90.20 | 39.59 | 5.75 | 68.29 | 61.96 | 60.59 | 39.74 |

norm of the magnitude to measure the importance of different channels in key cache.

$$M_{n,d} = \big\|\mathbf{K}[n,:,d]\big\|_p. \tag{1}$$

Given pruning ratio $\lambda$, We only keep $T = \lfloor(1 - \lambda)D\rfloor$ most important channels among the $D$ channels of each head: $I = \textbf{Top}_T(M, T)$ where $\|\cdot\|_p$ is the $l_p$ norm of each channel. $n \in [1, N]$ and $d \in [1, D]$ are indicators of heads and channels in key cache. $I \in (\mathbb{Z}^+)^{N \times T}$ stores the indicators of the top $T$ values in tensor $M$ per head.

In Table 1, we present the results of key cache pruning with various pruning ratios applied to the LLaMA-3-8B model. We utilize the $l_1$ and $l_2$ norms as criteria for evaluation, and validate performance using the LongBench benchmark (Bai et al., 2023). Compared to the baseline methods, H2O (Zhang et al., 2024c) and SnapKV (Li et al., 2024), both with a KV length of 512, we further prune the channels of the key cache. A 30% pruning ratio can maintain accuracy; however, increasing it to 40% results in significant performance degradation, especially for $l_1$ norm based pruning. The results of magnitude-based pruning support our assumption that the key cache is redundant in the channel dimension. These results also indicate the need for a better pruning metrics to achieve higher pruning ratios effectively.

## 3.2 QUERY-DRIVEN PRUNING

For each head, the attention scores are computed using the queries and keys, and then applied to the values. The formula for the attention for head $i$ is: $\text{Attention}(\mathbf{Q}_i, \mathbf{K}_i, \mathbf{V}_i) = \text{softmax}\big(\frac{\mathbf{Q}_i\mathbf{K}_i^T}{\sqrt{D}}\big)\mathbf{V}_i$, where $\mathbf{Q}_i, \mathbf{K}_i, \mathbf{V}_i \in \mathbb{R}^{S \times D}$. When one channel of $\mathbf{K}_i$ is pruned, the corresponding channel in $\mathbf{Q}_i$ will also be removed. We aim to find the optimal subset of channels to prune, denoted by the selection matrix $\mathbf{S} \in \{0,1\}^{D \times D}$, where $\mathbf{S}$ is a diagonal matrix with binary entries (1 for keeping a channel, 0 for pruning it). To better maintain the performance after pruning the channels, we minimize the Frobenius norm of the difference between the original and pruned attention weights: $\min_{\mathbf{S}} \|\mathbf{Q}_i\mathbf{K}_i^T - \mathbf{Q}_i\mathbf{S}(\mathbf{K}_i\mathbf{S})^T\|_F$. Given a pruning ratio $\lambda$, it can further expanded as:

$$\min_{\mathbf{S}} \quad \big\|\mathbf{Q}_i\mathbf{K}_i^T - \mathbf{Q}_i\mathbf{S}\mathbf{K}_i^T\big\|_F \tag{2}$$
$$\text{subject to} \quad \text{trace}(\mathbf{S}) = \lfloor(1 - \lambda)D\rfloor$$
$$\mathbf{S} = \text{diag}(s_1, s_2, \ldots, s_D), \text{ where } s_j \in \{0, 1\}$$

For simplicity, we use greedy algorithm to optimize $\mathbf{S}$. To achieve the pruning goal, we define a criterion for evaluating the importance of each channel and greedily select the channels with largest scores: $\text{Score}_i[j] = \big\|\mathbf{Q}_i[:,j]\mathbf{K}_i[:,j]^T\big\|_F$, $I_i = \textbf{Top}_T(\text{Score}_i, T)$. Here's a detailed explanation of why it optimizes the selection matrix. The $\text{score}_i[j]$ measures the magnitude of the interaction between the query and key vectors for channel $j$ in each head $i$. By selecting channels with the highest interaction magnitudes, we aim to retain the most significant contributions to the attention mechanism. This criterion ensures that the selected channels preserve the primary information flow in the attention computation, thereby minimizing the loss of important information.

**Observation Window.** Following SnapKV (Li et al., 2024), to reduce the computation cost, we only use the last $S_{obs}$ window to calculate the score as the last window of input sequence recognizes highly similar attention pattern with generation: $\|\mathbf{Q}_i[-S_{\text{obs}}:,j]\mathbf{K}_i[:,j]^T\|_F$.

Table 2: Performance comparison of key cache pruning on LLaMA-3-(8B/70B)-Instruct on Long-Bench. THINK ($\lambda$) indicates we prune the key cache channels with a pruning ratio of $\lambda$.

| Method | Single-Document QA | | | Multi-Document QA | | | Summarization | | | Few-shot Learning | | | Synthetic | | Code | | Avg. |
|---|---|---|---|---|---|---|---|---|---|---|---|---|---|---|---|---|---|
| | NrtvQA | Qasper | MF-en | HotpotQA | 2WikiMQA | Musique | GovReport | QMSum | MultiNews | TREC | TriviaQA | SAMSum | PCount | PRe | Lcc | RB-P | |
| *LLaMA-3-8B-Instruct, KV-size Full* | | | | | | | | | | | | | | | | | |
| ALL KV | 25.56 | 32.27 | 39.71 | 43.56 | 35.09 | 21.18 | 28.71 | 23.26 | 26.64 | 73.50 | 90.48 | 42.33 | 4.80 | 69.25 | 59.29 | 54.05 | 41.86 |
| *LLaMA-3-8B-Instruct, KV-size 128* | | | | | | | | | | | | | | | | | |
| H2O | 22.12 | 13.20 | 31.61 | 37.79 | 32.71 | 18.45 | 20.32 | 22.02 | 21.10 | 38.50 | 87.75 | 39.14 | 5.83 | 69.50 | 55.06 | 50.97 | 35.38 |
| +THINK (0.4) | 22.85 | 14.55 | 29.49 | 38.63 | 30.84 | 18.90 | 20.12 | 21.96 | 20.68 | 38.50 | 86.38 | 38.40 | 5.50 | 69.17 | 57.93 | 56.12 | **35.63** |
| +THINK (0.5) | 23.47 | 14.06 | 28.67 | 38.35 | 30.21 | 17.87 | 19.69 | 21.94 | 19.95 | 38.50 | 87.14 | 38.07 | 4.92 | 69.50 | 57.99 | 56.66 | 35.44 |
| SnapKV | 21.19 | 13.55 | 32.64 | 38.75 | 29.64 | 18.73 | 18.98 | 21.62 | 20.26 | 45.00 | 88.36 | 37.64 | 5.13 | 68.85 | 55.84 | 51.82 | 35.50 |
| +THINK (0.4) | 22.11 | 14.67 | 32.49 | 36.25 | 28.63 | 18.80 | 18.93 | 21.49 | 20.14 | 44.50 | 88.11 | 38.32 | 5.75 | 69.17 | 58.21 | 55.89 | **35.84** |
| +THINK (0.5) | 21.79 | 14.73 | 32.03 | 37.52 | 27.86 | 18.28 | 18.50 | 21.52 | 19.71 | 43.50 | 86.00 | 38.35 | 5.59 | 69.50 | 57.96 | 56.96 | 35.61 |
| *LLaMA-3-8B-Instruct, KV-size 512* | | | | | | | | | | | | | | | | | |
| H2O | 23.52 | 17.93 | 34.68 | 42.11 | 33.52 | 19.92 | 22.11 | 22.56 | 23.82 | 41.00 | 90.46 | 40.20 | 5.87 | 69.50 | 56.71 | 51.69 | 37.23 |
| +THINK (0.4) | 23.76 | 17.80 | 33.80 | 40.39 | 30.70 | 19.09 | 21.82 | 22.51 | 23.78 | 41.00 | 90.16 | 40.67 | 5.15 | 69.25 | 60.77 | 57.58 | **37.39** |
| +THINK (0.5) | 24.17 | 16.96 | 35.76 | 39.47 | 30.29 | 18.67 | 21.39 | 22.59 | 23.06 | 41.00 | 89.81 | 40.35 | 5.23 | 69.33 | 60.20 | 58.34 | 37.29 |
| +THINK (0.6) | 23.40 | 14.83 | 32.62 | 38.47 | 30.97 | 19.81 | 20.80 | 22.04 | 21.60 | 40.00 | 88.79 | 38.90 | 5.36 | 69.50 | 58.28 | 57.65 | 36.44 |
| SnapKV | 24.84 | 23.96 | 38.77 | 42.75 | 34.55 | 20.87 | 22.26 | 22.61 | 23.97 | 70.00 | 90.52 | 40.29 | 5.81 | 69.50 | 59.04 | 51.81 | 40.10 |
| +THINK (0.4) | 24.58 | 25.44 | 37.03 | 41.87 | 33.45 | 20.58 | 21.77 | 22.42 | 24.16 | 70.00 | 90.39 | 40.29 | 6.06 | 69.50 | 62.05 | 59.23 | **40.55** |
| +THINK (0.5) | 24.85 | 25.10 | 37.06 | 41.58 | 32.34 | 20.60 | 21.61 | 22.44 | 23.66 | 69.50 | 90.39 | 39.70 | 5.84 | 69.79 | 61.57 | 59.42 | 40.34 |
| +THINK (0.6) | 25.98 | 22.77 | 38.37 | 40.44 | 33.19 | 19.90 | 20.84 | 22.21 | 22.55 | 59.00 | 90.32 | 38.12 | 6.39 | 69.50 | 59.14 | 58.40 | 39.20 |
| *LLaMA-3-8B-Instruct, KV-size 1024* | | | | | | | | | | | | | | | | | |
| H2O | 25.62 | 22.16 | 36.81 | 41.01 | 33.53 | 19.41 | 23.28 | 22.65 | 25.41 | 46.50 | 90.82 | 41.78 | 5.79 | 69.25 | 59.69 | 55.50 | 38.70 |
| +THINK (0.4) | 25.52 | 21.93 | 37.17 | 41.56 | 31.22 | 20.17 | 22.89 | 22.95 | 25.21 | 47.00 | 90.74 | 41.34 | 5.57 | 69.50 | 62.58 | 58.67 | **39.00** |
| +THINK (0.5) | 25.41 | 22.19 | 37.64 | 40.92 | 31.27 | 18.66 | 22.17 | 22.22 | 24.84 | 46.50 | 90.34 | 40.59 | 5.20 | 69.50 | 61.71 | 57.99 | 38.57 |
| +THINK (0.6) | 24.06 | 17.80 | 37.85 | 38.63 | 29.98 | 19.40 | 21.41 | 22.32 | 23.42 | 44.50 | 90.16 | 39.43 | 5.84 | 69.50 | 58.31 | 58.73 | 37.58 |
| SnapKV | 24.62 | 25.99 | 37.64 | 43.84 | 34.99 | 20.00 | 24.28 | 22.39 | 25.63 | 72.5 | 90.56 | 40.41 | 5.36 | 69.25 | 60.57 | 56.11 | 40.88 |
| +THINK (0.4) | 24.88 | 27.72 | 38.60 | 43.16 | 32.44 | 20.67 | 24.21 | 22.79 | 25.56 | 71.50 | 90.45 | 40.94 | 5.93 | 69.50 | 62.77 | 59.45 | **41.29** |
| +THINK (0.5) | 24.82 | 27.26 | 39.66 | 42.82 | 32.09 | 19.56 | 23.52 | 22.48 | 25.34 | 71.50 | 90.43 | 40.74 | 5.20 | 69.50 | 62.46 | 59.75 | 41.07 |
| +THINK (0.6) | 24.46 | 27.35 | 38.22 | 41.96 | 31.64 | 20.18 | 21.89 | 22.83 | 23.68 | 70.00 | 90.19 | 38.69 | 6.10 | 69.50 | 58.87 | 59.26 | 40.30 |
| *LLaMA-3-8B-Instruct, KV-size 2048* | | | | | | | | | | | | | | | | | |
| H2O | 25.56 | 26.85 | 39.54 | 44.30 | 32.92 | 21.09 | 24.68 | 23.01 | 26.16 | 53.00 | 90.65 | 41.84 | 4.91 | 69.25 | 58.43 | 51.31 | 39.59 |
| +THINK (0.4) | 25.56 | 26.31 | 39.20 | 42.96 | 31.81 | 20.53 | 24.23 | 23.35 | 25.90 | 53.50 | 90.56 | 41.03 | 5.52 | 69.25 | 62.10 | 59.00 | **40.05** |
| +THINK (0.5) | 25.01 | 25.37 | 38.82 | 42.32 | 31.27 | 20.50 | 23.78 | 23.21 | 26.03 | 53.00 | 90.37 | 40.86 | 5.13 | 69.50 | 61.91 | 58.95 | 39.75 |
| +THINK (0.6) | 24.37 | 22.14 | 37.77 | 40.13 | 29.50 | 20.26 | 22.09 | 22.76 | 24.78 | 49.50 | 90.16 | 39.69 | 5.56 | 69.50 | 59.24 | 58.78 | 38.51 |
| SnapKV | 25.86 | 29.55 | 41.10 | 44.99 | 35.80 | 21.81 | 25.98 | 23.40 | 26.46 | 73.50 | 90.56 | 41.66 | 5.17 | 69.25 | 58.67 | 51.52 | 41.58 |
| +THINK (0.4) | 25.41 | 29.79 | 39.21 | 43.35 | 33.96 | 21.49 | 25.78 | 23.11 | 26.23 | 73.00 | 90.56 | 41.79 | 5.81 | 69.50 | 62.45 | 59.19 | **41.91** |
| +THINK (0.5) | 25.00 | 30.25 | 39.27 | 43.23 | 32.93 | 21.24 | 25.16 | 23.01 | 26.5 | 73.00 | 90.37 | 41.26 | 5.45 | 69.50 | 62.3 | 59.84 | 41.77 |
| +THINK (0.6) | 24.89 | 28.88 | 40.44 | 41.30 | 29.99 | 21.34 | 23.48 | 22.9 | 24.99 | 72.50 | 90.36 | 38.5 | 5.71 | 69.50 | 59.77 | 59.50 | 40.88 |
| *LLaMA-3-70B-Instruct, KV-size 128* | | | | | | | | | | | | | | | | | |
| SnapKV | 25.91 | 39.41 | 43.83 | 49.60 | 51.23 | 27.76 | 22.14 | 21.91 | 23.16 | 69.00 | 91.55 | 43.54 | 12.50 | 72.00 | 48.41 | 63.49 | **44.09** |
| +THINK (0.4) | 25.64 | 39.20 | 43.60 | 50.22 | 50.50 | 29.32 | 21.70 | 21.96 | 23.35 | 68.00 | 91.27 | 43.24 | 12.50 | 73.00 | 48.01 | 62.43 | 44.00 |
| +THINK (0.5) | 26.31 | 38.76 | 44.86 | 48.54 | 49.62 | 28.97 | 21.46 | 22.01 | 22.91 | 67.00 | 91.52 | 43.15 | 12.50 | 72.50 | 47.21 | 60.82 | 43.63 |
| *LLaMA-3-70B-Instruct, KV-size 512* | | | | | | | | | | | | | | | | | |
| SnapKV | 27.95 | 45.19 | 48.50 | 50.97 | 54.53 | 29.78 | 25.34 | 22.36 | 26.03 | 73.50 | 92.63 | 45.07 | 12.50 | 72.50 | 45.21 | 68.22 | 46.27 |
| +THINK (0.4) | 27.47 | 45.31 | 48.57 | 51.22 | 54.32 | 30.05 | 25.42 | 22.72 | 26.20 | 73.50 | 92.13 | 45.53 | 12.50 | 73.00 | 48.32 | 66.99 | **46.45** |
| +THINK (0.5) | 26.97 | 44.55 | 48.16 | 50.84 | 53.80 | 30.57 | 25.29 | 22.65 | 25.53 | 73.00 | 92.13 | 43.66 | 12.50 | 73.00 | 50.52 | 64.82 | 46.12 |
| *LLaMA-3-70B-Instruct, KV-size 1024* | | | | | | | | | | | | | | | | | |
| SnapKV | 26.80 | 46.21 | 49.93 | 51.70 | 54.71 | 29.86 | 27.61 | 22.43 | 27.15 | 73.50 | 92.38 | 46.18 | 12.50 | 72.50 | 42.84 | 69.89 | 46.64 |
| +THINK (0.4) | 27.04 | 46.01 | 50.13 | 51.96 | 54.36 | 29.87 | 27.74 | 22.78 | 27.07 | 73.50 | 91.88 | 46.35 | 12.50 | 73.00 | 45.05 | 67.87 | **46.69** |
| +THINK (0.5) | 27.62 | 46.22 | 48.97 | 51.79 | 53.39 | 30.47 | 27.45 | 23.05 | 26.57 | 73.50 | 91.88 | 43.99 | 12.50 | 72.50 | 47.41 | 66.84 | 46.51 |
| *LLaMA-3-70B-Instruct, KV-size 2048* | | | | | | | | | | | | | | | | | |
| SnapKV | 27.44 | 46.51 | 49.60 | 51.80 | 54.77 | 31.05 | 29.67 | 22.44 | 27.43 | 73.50 | 92.38 | 45.98 | 12.50 | 72.50 | 41.86 | 68.72 | **46.76** |
| +THINK (0.4) | 27.13 | 46.26 | 50.04 | 51.72 | 55.03 | 31.19 | 29.75 | 22.47 | 27.28 | 73.50 | 91.88 | 46.37 | 12.50 | 72.50 | 42.66 | 67.77 | 46.75 |
| +THINK (0.5) | 27.84 | 46.86 | 49.18 | 51.97 | 53.58 | 31.44 | 29.41 | 22.89 | 27.33 | 73.50 | 91.88 | 43.60 | 12.50 | 72.50 | 44.78 | 66.65 | 46.62 |

## 3.3 IMPLEMENTATION OF THINK

We opt not to prune the most recent tokens and newly generated keys. Consequently, our key-value (KV) cache has two categories: one subset consists of pruned keys with a reduced channel size, while the other at their original size. Additionally, we maintain a binary mask whose memory overhead is negligible to indicate which channels have been pruned. Figure 2 illustrates one implementation of THINK during the decoding stage. We first prune the query using the stored mask, ensuring that the query dimensionality and the pruned key cache remain consistent. The pruned query is then multiplied by the pruned key, while the unpruned query is applied to the unpruned key. Subsequently, the two outputs are concatenated. Our method removes unimportant Key cache channels to get real memory reduction.

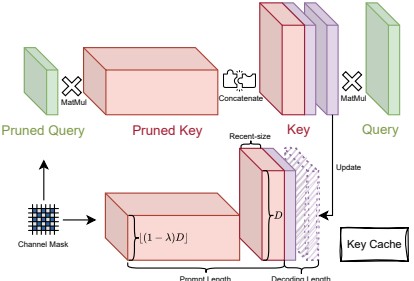

Figure 2: Implementation during decoding.

Table 3: Performance comparison of key cache pruning on Mistral-7B-Instruct-v0.2 on LongBench. THINK ($\lambda$) indicates we prune the key cache channels with a pruning ratio of $\lambda$.

| Method | Single-Document QA | | | Multi-Document QA | | | Summarization | | | Few-shot Learning | | | Synthetic | | Code | | Avg. |
|---|---|---|---|---|---|---|---|---|---|---|---|---|---|---|---|---|---|
| | NrtvQA | Qasper | MF-en | HotpotQA | 2WikiMQA | Musique | GovReport | QMSum | MultiNews | TREC | TriviaQA | SAMSum | PCount | PRe | Lcc | RB-P | |
| Mistral-7B-Instruct-v0.2, KV-size Full | | | | | | | | | | | | | | | | | |
| ALL KV | 26.63 | 32.99 | 49.34 | 42.77 | 27.35 | 18.77 | 32.87 | 24.24 | 27.10 | 71.00 | 86.23 | 42.96 | 2.75 | 86.98 | 56.93 | 54.49 | 42.71 |
| Mistral-7B-Instruct-v0.2, KV-size 128 | | | | | | | | | | | | | | | | | |
| H2O | 21.21 | 21.81 | 38.87 | 30.42 | 20.36 | 12.30 | 20.58 | 22.61 | 22.10 | 39.00 | 82.37 | 40.44 | 2.90 | 79.56 | 51.22 | 48.38 | 34.63 |
| +THINK (0.4) | 21.17 | 21.90 | 39.29 | 29.92 | 20.99 | 12.30 | 20.84 | 22.91 | 21.92 | 39.00 | 82.70 | 40.35 | 2.97 | 79.21 | 51.19 | 48.32 | **34.69** |
| +THINK (0.5) | 21.67 | 21.80 | 39.48 | 28.74 | 20.65 | 13.14 | 20.57 | 22.83 | 21.78 | 39.00 | 82.54 | 40.12 | 3.61 | 78.39 | 50.27 | 48.4 | 34.56 |
| +THINK (0.6) | 21.04 | 21.30 | 39.56 | 28.68 | 21.29 | 13.97 | 20.13 | 22.52 | 21.81 | 39.50 | 82.05 | 39.14 | 4.16 | 74.23 | 49.83 | 47.67 | 34.18 |
| SnapKV | 19.17 | 21.40 | 42.93 | 36.76 | 22.44 | 15.86 | 19.16 | 21.84 | 21.55 | 47.50 | 84.15 | 40.24 | 2.30 | 68.26 | 52.31 | 48.80 | **35.29** |
| +THINK (0.4) | 20.52 | 21.00 | 42.65 | 37.58 | 22.09 | 15.23 | 19.29 | 22.01 | 21.22 | 47.00 | 83.85 | 39.64 | 3.20 | 67.45 | 51.48 | 48.31 | 35.16 |
| +THINK (0.5) | 20.67 | 20.60 | 43.37 | 37.27 | 21.58 | 15.66 | 19.06 | 21.79 | 21.02 | 47.00 | 83.38 | 39.77 | 3.65 | 67.06 | 50.80 | 48.35 | 35.06 |
| +THINK (0.6) | 21.25 | 20.82 | 44.20 | 36.21 | 21.68 | 16.47 | 19.05 | 21.99 | 20.73 | 45.00 | 83.81 | 38.79 | 4.19 | 66.90 | 49.99 | 47.61 | 34.92 |
| Mistral-7B-Instruct-v0.2, KV-size 512 | | | | | | | | | | | | | | | | | |
| H2O | 21.83 | 26.00 | 44.69 | 32.46 | 23.05 | 14.69 | 23.53 | 23.06 | 24.59 | 42.00 | 85.22 | 41.49 | 3.40 | 86.20 | 54.78 | 51.09 | 37.38 |
| +THINK (0.4) | 21.58 | 26.15 | 44.49 | 32.73 | 23.99 | 15.09 | 23.56 | 23.28 | 24.45 | 42.00 | 85.58 | 42.58 | 3.18 | 85.7 | 54.39 | 51.15 | **37.49** |
| +THINK (0.5) | 22.76 | 25.74 | 44.61 | 31.74 | 23.25 | 13.91 | 23.31 | 23.13 | 24.34 | 41.00 | 85.39 | 41.85 | 2.82 | 84.36 | 54.69 | 50.88 | 37.11 |
| +THINK (0.6) | 22.91 | 25.57 | 44.04 | 29.48 | 22.88 | 13.67 | 23.31 | 22.64 | 24.10 | 41.00 | 85.31 | 41.15 | 2.98 | 82.34 | 53.70 | 50.25 | 36.58 |
| SnapKV | 24.44 | 27.81 | 48.98 | 39.46 | 25.25 | 16.98 | 23.70 | 22.96 | 24.37 | 67.00 | 85.88 | 41.26 | 2.78 | 86.56 | 56.46 | 53.41 | 40.46 |
| +THINK (0.4) | 24.27 | 28.46 | 49.26 | 38.13 | 24.22 | 16.92 | 23.59 | 23.70 | 24.46 | 67.50 | 85.9 | 42.51 | 2.92 | 85.32 | 55.89 | 53.35 | 40.40 |
| +THINK (0.5) | 24.56 | 29.22 | 48.59 | 37.70 | 24.27 | 17.39 | 23.68 | 23.65 | 24.58 | 67.50 | 86.05 | 42.01 | 3.07 | 86.30 | 56.49 | 53.29 | **40.52** |
| +THINK (0.6) | 24.07 | 28.27 | 49.10 | 38.65 | 24.31 | 17.52 | 23.16 | 23.51 | 24.23 | 67.00 | 86.33 | 40.78 | 3.69 | 83.74 | 54.94 | 52.23 | 40.10 |
| Mistral-7B-Instruct-v0.2, KV-size 1024 | | | | | | | | | | | | | | | | | |
| H2O | 23.67 | 28.55 | 46.40 | 36.99 | 24.82 | 15.02 | 25.21 | 23.04 | 25.77 | 46.00 | 85.93 | 41.98 | 3.24 | 86.57 | 56.40 | 52.75 | **38.90** |
| +THINK (0.4) | 23.97 | 28.91 | 45.84 | 35.78 | 24.88 | 14.55 | 25.11 | 23.35 | 25.83 | 45.50 | 86.11 | 42.44 | 3.23 | 84.82 | 56.21 | 53.02 | 38.72 |
| +THINK (0.5) | 23.89 | 28.40 | 46.60 | 35.57 | 24.26 | 14.78 | 24.98 | 23.31 | 25.68 | 44.50 | 86.16 | 42.72 | 3.38 | 83.20 | 55.88 | 52.63 | 38.50 |
| +THINK (0.6) | 23.87 | 27.76 | 46.25 | 35.28 | 24.38 | 14.74 | 24.35 | 23.35 | 25.50 | 44.50 | 85.38 | 41.37 | 3.34 | 81.42 | 55.21 | 51.89 | 38.04 |
| SnapKV | 25.47 | 29.57 | 49.33 | 40.90 | 25.53 | 19.01 | 25.94 | 23.89 | 26.21 | 69.50 | 86.48 | 42.10 | 2.98 | 88.56 | 57.19 | 53.60 | **41.64** |
| +THINK (0.4) | 25.22 | 30.48 | 48.58 | 41.11 | 25.28 | 18.99 | 25.91 | 24.00 | 26.13 | 70.00 | 86.64 | 43.35 | 2.98 | 86.3 | 56.71 | 54.19 | 41.62 |
| +THINK (0.5) | 25.63 | 30.08 | 49.41 | 40.59 | 25.13 | 19.58 | 25.47 | 24.23 | 25.92 | 69.5 | 86.67 | 42.31 | 2.74 | 84.78 | 57.43 | 53.59 | 41.44 |
| +THINK (0.6) | 24.69 | 29.3 | 48.90 | 40.44 | 25.33 | 19.58 | 25.23 | 23.6 | 25.25 | 69.00 | 86.85 | 40.86 | 3.19 | 83.70 | 56.3 | 53.30 | 40.97 |
| Mistral-7B-Instruct-v0.2, KV-size 2048 | | | | | | | | | | | | | | | | | |
| H2O | 25.76 | 31.10 | 49.06 | 40.38 | 26.43 | 16.78 | 27.17 | 23.64 | 26.69 | 55.00 | 86.35 | 42.48 | 2.72 | 86.64 | 56.98 | 53.91 | **40.69** |
| +THINK (0.4) | 25.40 | 30.80 | 48.45 | 39.64 | 26.08 | 16.82 | 27.12 | 23.79 | 26.65 | 53.50 | 86.39 | 43.03 | 3.29 | 86.39 | 56.61 | 53.60 | 40.47 |
| +THINK (0.5) | 25.68 | 31.24 | 48.69 | 39.65 | 25.84 | 16.72 | 26.69 | 23.57 | 26.78 | 52.00 | 86.74 | 42.85 | 4.01 | 83.46 | 57.12 | 53.67 | 40.29 |
| +THINK (0.6) | 25.83 | 31.00 | 48.23 | 38.58 | 25.71 | 16.54 | 26.51 | 23.81 | 26.28 | 50.50 | 86.57 | 42.05 | 3.36 | 82.49 | 56.04 | 52.67 | 39.76 |
| SnapKV | 25.89 | 32.56 | 48.55 | 41.68 | 27.24 | 18.75 | 28.90 | 24.47 | 26.63 | 70.00 | 86.27 | 42.57 | 3.09 | 86.93 | 57.44 | 53.83 | 42.18 |
| +THINK (0.4) | 25.77 | 32.67 | 48.70 | 41.06 | 27.07 | 19.14 | 28.91 | 24.37 | 26.88 | 70.00 | 86.37 | 42.75 | 3.61 | 87.38 | 57.21 | 54.44 | **42.27** |
| +THINK (0.5) | 26.44 | 32.94 | 49.02 | 40.86 | 26.84 | 19.49 | 28.46 | 24.51 | 26.72 | 70.00 | 86.50 | 41.75 | 2.78 | 84.70 | 56.47 | 54.15 | 41.98 |
| +THINK (0.6) | 26.00 | 32.53 | 48.73 | 40.95 | 26.77 | 18.92 | 27.40 | 23.97 | 26.37 | 70.00 | 86.45 | 41.12 | 3.31 | 82.24 | 56.01 | 53.53 | 41.52 |

# 4 EXPERIMENT RESULTS

In this section, we conduct comprehensive experiments to evaluate the effectiveness of THINK on performance and memory reduction. In the experiments, we prune Key cache channels by default, the value cache results are in Table 8 in Appendix D.

## 4.1 SETTINGS

**Benchmark Datasets.** We evaluate our proposed method against state-of-the-art KV cache compression methods on two widely recognized benchmarks: LongBench and Needle-in-a-Haystack. *LongBench* (Bai et al., 2023) is designed to comprehensively assess the long context understanding capabilities of LLMs. It includes 17 datasets covering six different tasks: single-document QA, multi-document QA, summarization, few-shot learning, synthetic tasks, and code completion. The average input length of LongBench is 6,711 words, which necessitates reducing the KV cache to lower memory usage for inference. *Needle-in-a-Haystack* (Kamradt, 2023) is a recently developed benchmark that tests a model's ability to accurately locate a small but crucial piece of information (the "needle") embedded within a lengthy document (the "haystack"). The random positioning of the needle in this challenge serves as a critical test to determine whether KV cache compression methods can retain essential information without loss of accuracy.

**Baseline Approaches.** The baseline methods in our evaluations include Heavy Hitter Oracle (H2O), SnapKV and KIVI, all of which are the state-of-the-art KV cache compression methods but use different strategies. **H2O** (Zhang et al., 2024c) is designed to reduce memory usage by dynamically managing the balance between recent tokens and Heavy Hitter (H2) tokens. H2 tokens represent a small set of tokens that contribute most of the value when computing attention scores. **SnapKV** (Li et al., 2024) introduces an automated compression mechanism that selects clustered,

Table 4: Performance evaluation of combining THINK with KIVI Liu et al. (2024b) on LongBench. THINK (0.4) indicates we prune the key cache channels with a pruning ratio of $\lambda = 0.4$.

| Method | Bit | Single-Document QA | | | Multi-Document QA | | | Summarization | | | Few-shot Learning | | | Synthetic | | Code | | Avg. |
| | | NrtvQA | Qasper | MF-en | HotpotQA | 2WikiMQA | Musique | GovReport | QMSum | MultiNews | TREC | TriviaQA | SAMSum | PCount | PRe | Lcc | RB-P | |
| KIVI | 2/2 | 19.47 | 18.62 | 30.28 | 29.42 | 25.00 | 10.30 | 21.34 | 20.51 | 25.10 | 63.00 | 85.04 | 40.16 | 4.00 | 8.00 | 58.04 | 52.48 | 31.92 |
| +THINK (0.4) | 2/2 | 19.46 | 19.01 | 30.52 | 28.79 | 25.78 | 9.53 | 22.11 | 20.66 | 25.73 | 63.00 | 84.62 | 41.54 | 3.50 | 7.00 | 56.51 | 48.92 | 31.77 |

important KV positions for each attention head, optimizing the KV cache without sacrificing performance. **KIVI** (Liu et al., 2024b) reduces memory overhead by quantizing the KV cache into lower-precision formats, significantly lowering the memory cost while preserving model accuracy.

**Implementation Details.** In this paper, we use LLaMA-2-7B-chat, LLaMA-3-8B-Instruct, LLaMA-3-70B-Instruct (Meta, 2024) and Mistral-7B-Instruct-v0.2 (Jiang et al., 2023) as the backbone LLMs, both accessible via HuggingFace (Wolf et al., 2020). Our THINK aims to prune channels of the key cache, which is agnostic to KV cache compression methods. If there is no other statement, we prune the key cache by default. All the experiments are conducted using NVIDIA A100 GPUs. To ensure a fair comparison between KV cache compression methods and their integration with THINK, we applied consistent hyperparameters across both settings. For instance, when comparing SnapKV and SnapKV integrated with THINK, we used a maximum pooling kernel size of 7 and an observation window size of 32, maintaining the same KV-size for both configurations. We compress the Key cache starting from the prefilling stage.

## 4.2 RESULTS ON LONGBENCH

Tables 2 and 3 present the results of KV compression methods and their integration with our proposed channel pruning technique for the key cache (THINK) across three different base LLMs, evaluated at various KV-sizes on the LongBench benchmark. The pruning ratio of $\lambda = 0.4$ indicate that 40% of key cache channels are removed ,resulting in a 20% reduction in the total KV cache memory footprint. The following observations can be drawn: (1) Our method successfully prunes the channels of the key cache after the KV cache has been compressed using H2O and SnapKV. For the LLaMA-3-8B-Instruct base model, our approach reduces memory usage while slightly improving performance for both H2O and SnapKV. For the Mistral-7B-Instruct-v0.2 base model, our method similarly reduces memory usage, with only a minor performance drop in some cases. For the larger LLaMA-3-70B base model, our method achieves comparable or superior performance after pruning 40% of the key cache channels, compared to the SnapKV baselines. (2) When Comparing SnapKV or H2O integrated with THINK in Table 2 to SnapKV or H2O integrated with $l_1$ or $l_2$ norm in Table 1, our query-driven channel pruning approach demonstrates superior performance when the pruning ratio of $\lambda = 0.4$. (3) Lower pruning ratios generally result in better performance compared to higher pruning ratios. (4) As the KV-size increases from 128 to 2048, the performance of our channel pruning method improves. Notably, with a KV-size of 2048 and a pruning ratio of 0.4, our method even surpasses the performance of LLaMA-3-8B-Instruct with a full KV cache. These findings suggest that our method is agnostic to the underlying KV cache compression techniques and can further enhance both performance and memory efficiency. Moreover, query-driven channel pruning proves more effective than $l_1$ and $l_2$ norm-based methods for channel pruning in LLMs. Experiment results of applying THINK directly to vanilla models are in Table 9 which further validate the effectiveness of our method.

We further validate the efficacy of our method by applying it to the KV cache quantization technique KIVI (Liu et al., 2024b), as shown in Table 4. First, we prune 40% of the key cache channels, followed by quantization of the remaining channels into 2-bit (implementation details provided in Appendix C.1). Compared to the standard KIVI method, our approach reduces KV cache memory by 20%, with minimal performance degradation.

## 4.3 RESULTS ON NEEDLE-IN-A-HAYSTACK

Table 5 presents the results of the Needle-in-a-Haystack test, using the SnapKV (Li et al., 2024) approach with varying KV-sizes, ranging from 128 to 2048. With a modest pruning ratio of

Table 5: Needles-in-a-Haystack Test Results

| Model | Method | $\lambda$ | KV-size | | | |
|-------|--------|-----------|-----|-----|------|------|
| | | | 128 | 512 | 1024 | 2048 |
| LLaMA3-8B-Instruct | SnapKV | 0.0 | 79.6 | 90.2 | 91.2 | 91.7 |
| | SnapKV+THINK | 0.4 | 79.6 | 90.3 | 91.2 | 91.7 |
| | SnapKV+THINK | 0.5 | 77.4 | 89.6 | 91.0 | 91.7 |
| Mistral-7B-Instruct-v0.2 | SnapKV | 0.0 | 77.8 | 89.5 | 90.4 | 90.8 |
| | SnapKV+THINK | 0.4 | 78.6 | 90.1 | 90.6 | 90.9 |
| | SnapKV+THINK | 0.5 | 78.1 | 90.1 | 90.8 | 91.1 |
| | SnapKV+THINK | 0.6 | 75.9 | 89.2 | 90.6 | 91.1 |

Table 6: Performance comparison of key cache pruning with varying recent-sizes.

| Recent-Size | Single-Document QA | | | Multi-Document QA | | | Summarization | | | Few-shot Learning | | | Synthetic | | Code | | Avg. |
|---|---|---|---|---|---|---|---|---|---|---|---|---|---|---|---|---|---|
| | NrtvQA | Qasper | MF-en | HotpotQA | 2WikiMQA | Musique | GovReport | QMSum | MultiNews | TREC | TriviaQA | SAMSum | PCount | PRe | Lcc | RB-P | |
| | | | | | | | H2O + THINK ($\lambda = 0.4$) | | | | | | | | | | |
| 0 | 24.40 | 27.50 | 45.42 | 35.17 | 24.45 | 13.02 | 27.65 | 23.88 | 26.86 | 53.50 | 86.06 | 41.73 | 3.01 | 83.42 | 55.12 | 51.32 | 38.91 |
| 32 | 25.40 | 30.80 | 48.45 | 39.64 | 26.08 | 16.82 | 27.12 | 23.79 | 26.65 | 53.50 | 86.39 | 43.03 | 3.29 | 86.39 | 56.61 | 53.60 | 40.47 |
| 128 | 25.69 | 30.93 | 48.32 | 39.63 | 26.08 | 16.82 | 27.18 | 23.92 | 26.62 | 53.50 | 86.39 | 42.96 | 3.29 | 86.39 | 56.77 | 53.60 | **40.51** |
| | | | | | | | SnapKV + THINK ($\lambda = 0.4$) | | | | | | | | | | |
| 0 | 24.94 | 28.58 | 45.78 | 39.59 | 25.40 | 15.92 | 29.50 | 24.05 | 26.72 | 70.00 | 85.60 | 41.38 | 2.97 | 84.00 | 55.27 | 52.39 | 40.76 |
| 32 | 25.77 | 32.67 | 48.70 | 41.06 | 27.07 | 19.14 | 28.91 | 24.37 | 26.88 | 70.00 | 86.37 | 42.75 | 3.61 | 87.38 | 57.21 | 54.44 | **42.27** |
| 128 | 25.75 | 32.49 | 48.61 | 41.01 | 27.18 | 19.14 | 28.79 | 24.64 | 26.77 | 70.00 | 86.37 | 42.77 | 3.61 | 87.13 | 57.19 | 54.39 | 42.24 |

$\lambda = 0.4$, THINK consistently outperforms or matches the accuracy of the original SnapKV across both LLaMA-3 and Mistral models, regardless of KV-size. These comparisons demonstrate that the proposed query-driven channel pruning method effectively retains informative channels while discarding noisy ones. However, when the pruning ratio increases to $\lambda \geq 0.5$, we observe a drop in accuracy with smaller KV-sizes, particularly for 128 and 512, across both LLaMA-3 and Mistral models. Despite this, THINK achieves comparable performance with SnapKV when the KV-size is larger(*i.e.*, 1024 and 2048). Intuitively, a larger pruning ratio with a smaller KV-size may lead to the loss of more critical information compared to scenarios with a larger KV-size. In addition, the performance on larger KV-sizes suggests that THINK is robust for long-context tasks.

Figure 6 (a)-(d) (in Appendix B) visualize the Needle-in-a-Haystack test accuracy across varying token lengths and depths. The KV-sizes are set to 128 and 1024, with pruning ratios of $\lambda = 0.4$ and $\lambda = 0.5$, respectively. THINK preserves the retrieval capabilities of SnapKV, although there are minor numerical differences in overall accuracy (e.g., 77.8 vs. 78.6 and 90.4 vs. 90.6). THINK matches SnapKV in accuracy for the majority of token limits and depths, demonstrating consistency in performance. Furthermore, THINK successfully retrieves certain "needles" that SnapKV fails to capture, resulting in improved overall accuracy. These visualizations highlight the robustness of THINK from a fine-grained perspective, illustrating its capacity to enhance the original approach.

## 4.4 ABLATION STUDIES

**Impact of Different Recent Sizes.** Preserving the most recent KV embeddings (Zhang et al., 2024c; Li et al., 2024) is important for maintaining the performance of LLMs after KV cache compression. However, a tradeoff exists: increasing the recent-size allows more information to be retained, but also increases the cache size. To assess its impact, we evaluate the performance of three recent-size configurations, namely 0, 32 and 128, on LongBench, using Mistral-7B-Instruct-v0.2 as the baseline model. The results are summarized in Table 6. As observed, a recent-size of 32 yields superior performance compared to 0, as indicated by the averaged score on LongBench, demonstrating the importance of retaining the most recent KVs. On the other hand, the performance difference between recent-sizes of 32 and 128 is negligible, suggesting that retaining the most recent 32 KVs is sufficient to maintain optimal performance.

**Performance Comparison Under the Same Memory Usage.** To ensure a fair comparison, we adjust the KV-size of H2O or SnapKV to match the memory usage of H2O with THINK or SnapKV with THINK on Mistral-7B-Instruct-v0.2. For example, the KV-size of H2O with THINK is set to 128. Due to channel pruning applied to the key cache, the memory consumption of H2O with

Table 7: Performance comparison of key cache pruning with the same memory consumption.

| Methods | Memory(M) | Single-Document QA | | | Multi-Document QA | | | Summarization | | | Few-shot Learning | | | Synthetic | | Code | | Avg. |
|---|---|---|---|---|---|---|---|---|---|---|---|---|---|---|---|---|---|---|
| | | NrtvQA | Qasper | MF-en | HotpotQA | 2WikiMQA | Musique | GovReport | QMSum | MultiNews | TREC | TriviaQA | SAMSum | PCount | PRe | Lcc | RB-P | |
| | | | | | | | | H2O | | | | | | | | | | |
| Vanilla | 54.5 | 21.29 | 20.69 | 37.66 | 28.65 | 21.08 | 14.01 | 20.20 | 22.11 | 21.33 | 38.50 | 82.55 | 39.87 | 3.66 | 78.14 | 50.32 | 48.54 | 34.29 |
| THINK | 54.4 | 21.17 | 21.90 | 39.29 | 29.92 | 20.99 | 12.30 | 20.84 | 22.91 | 21.92 | 39.00 | 82.70 | 40.35 | 2.97 | 79.21 | 51.19 | 48.32 | **34.69** |
| Vanilla | 208.0 | 22.13 | 23.83 | 43.24 | 30.92 | 23.36 | 14.56 | 22.92 | 22.77 | 24.23 | 41.50 | 85.04 | 41.26 | 3.02 | 86.03 | 54.91 | 50.50 | 36.89 |
| THINK | 208.0 | 21.58 | 26.15 | 44.49 | 32.73 | 23.99 | 15.09 | 23.56 | 23.28 | 24.45 | 42.00 | 85.58 | 42.58 | 3.18 | 85.7 | 54.39 | 51.15 | **37.49** |
| Vanilla | 413.0 | 22.90 | 28.45 | 46.16 | 35.57 | 23.86 | 13.74 | 24.90 | 23.19 | 25.77 | 44.50 | 85.54 | 41.97 | 3.22 | 85.82 | 55.96 | 52.33 | 38.37 |
| THINK | 412.8 | 23.97 | 28.91 | 45.84 | 35.78 | 24.88 | 14.55 | 25.11 | 23.35 | 25.83 | 45.50 | 86.11 | 42.44 | 3.23 | 84.82 | 56.21 | 53.02 | **38.72** |
| Vanilla | 822.5 | 25.51 | 30.23 | 48.23 | 39.72 | 25.56 | 16.75 | 26.98 | 23.81 | 26.47 | 50.50 | 86.43 | 42.09 | 2.78 | 85.57 | 57.4 | 53.42 | 40.09 |
| THINK | 822.4 | 25.40 | 30.80 | 48.45 | 39.64 | 26.08 | 16.82 | 27.12 | 23.79 | 26.65 | 53.50 | 86.39 | 43.03 | 3.29 | 86.39 | 56.61 | 53.60 | **40.47** |
| | | | | | | | | SnapKV | | | | | | | | | | |
| Vanilla | 54.5 | 19.25 | 19.95 | 42.80 | 35.88 | 21.96 | 14.59 | 18.76 | 21.71 | 20.70 | 46.00 | 84.12 | 39.43 | 2.59 | 65.36 | 51.39 | 47.81 | 34.52 |
| THINK | 54.4 | 20.52 | 21.00 | 42.65 | 37.58 | 22.09 | 15.23 | 19.29 | 22.01 | 21.22 | 47.00 | 83.85 | 39.64 | 3.20 | 67.45 | 51.48 | 48.31 | **35.16** |
| Vanilla | 208.0 | 23.31 | 27.45 | 48.85 | 38.77 | 23.93 | 16.50 | 23.44 | 23.63 | 24.13 | 66.00 | 86.05 | 41.00 | 2.62 | 87.01 | 56.13 | 52.60 | 40.09 |
| THINK | 208.0 | 24.27 | 28.46 | 49.26 | 38.13 | 24.22 | 16.92 | 23.59 | 23.70 | 24.46 | 67.50 | 85.90 | 42.51 | 2.92 | 85.32 | 55.89 | 53.35 | **40.40** |
| Vanilla | 413.0 | 24.24 | 29.53 | 49.13 | 40.48 | 25.05 | 18.74 | 25.46 | 23.64 | 25.60 | 68.00 | 86.14 | 41.42 | 3.03 | 88.55 | 57.08 | 53.86 | 41.25 |
| THINK | 412.8 | 25.22 | 30.48 | 48.58 | 41.11 | 25.28 | 18.99 | 25.91 | 24.00 | 26.13 | 70.00 | 86.64 | 43.35 | 2.98 | 86.30 | 56.71 | 54.19 | **41.62** |
| Vanilla | 822.5 | 24.84 | 31.90 | 48.16 | 41.32 | 26.77 | 19.49 | 28.23 | 24.63 | 26.41 | 70.00 | 86.32 | 41.83 | 2.91 | 88.06 | 56.98 | 53.74 | 41.97 |
| THINK | 822.4 | 25.77 | 32.67 | 48.7 | 41.06 | 27.07 | 19.14 | 28.91 | 24.37 | 26.88 | 70.00 | 86.37 | 42.75 | 3.61 | 87.38 | 57.21 | 54.44 | **42.27** |

Figure 3: (a) presents the performance comparison with token eviction methods under identical memory usage for Mistral-7B-Instruct-v0.2, while (b) illustrates the memory usage comparison with the KV cache quantization method KIVI across different batch sizes for LLaMA-2-7B-chat. THINK $(0.4)$ indicates we prune the key cache channels with a pruning ratio of $\lambda = 0.4$.

THINK at a KV-size of $128$ is lower than that of H2O at the same KV-size. Consequently, the KV-size of H2O is adjusted from $128$ to $109$ to equalize memory usage. Table 7 and Figure 3a present the results of these comparisons on the LongBench benchmark. The results demonstrate that H2O or SnapKV combined with THINK consistently outperforms their counterparts without THINK while maintaining the same memory footprint. This highlights the effectiveness of integrating query-driven channel pruning with KV cache compression methods, enabling more efficient memory utilization and improved compression of the KV cache.

**Memory Usage Comparison.** To evaluate the efficiency of THINK, we follow the methodology used in KIVI (Liu et al., 2024b). We generate synthetic workloads with an input prompt length of 160 and an output length of 338. The peak memory usage is reported for the vanilla FP16 baseline, KIVI, and KIVI combined with THINK $(0.4)$ for LLaMA-2-7B-chat. As in Figure 3b, the memory savings from our method become increasingly evident as the batch size grows, in both the KIVI $2/2$ and KIVI $4/4$ configurations. Compared to the baseline model, our approach achieves over a $5\times$ (from $4\times$ with KIVI alone) increase in batch size while maintaining the same memory footprint when integrated with KIVI. Model weights and KV cache are the primary memory components accessed during generation. By effectively reducing the memory footprint of the KV cache, our method alleviates the memory bottleneck, enabling faster generation speeds as shown in Table 10.

**Pruning Channels of Both Key and Value Cache.** In this part, we explore the impact of pruning channels in the value cache (Appendix D). Specifically, for KV cache compression methods, we apply different pruning ratios to the channels of the key and value caches. Table 8 presents the results with LLaMA-3-8B-Instruct and Mistral-7B-Instruct-v0.2 on the LongBench benchmark.

When evaluating the base model LLaMA-3-8B-Instruct, H2O or SnapKV with both key and value channel pruning perform comparably to H2O or SnapKV without pruning. In certain instances, the models with key and value channel pruning even outperform their non-pruned counterparts. For the base model Mistral-7B-Instruct-v0.2, pruning the value cache channels leads to a slight performance drop. This aligns with the observations in Figure 4, where the Key cache shows highly unbalanced magnitudes along the channel dimension, while the Value cache exhibits more uniform magnitudes. This suggests that the Value cache has less channel sparsity, making it harder to identify redundant channels for pruning. Nevertheless, pruning Value cache channels still contributes to further memory reduction in the KV cache.

## 5    RELATED WORK

In scenarios involving long contexts, the key-value (KV) cache poses the most significant computational and memory burden within the attention mechanism of large language models. Reducing the KV cache is therefore a key priority for optimizing deployment efficiency. To address this challenge, system-level optimizations, such as FlashAttention (Dao, 2023) and PagedAttention (Kwon et al., 2023), have been developed to tackle this challenge. In parallel, algorithm-level optimizations are also being explored to further enhance efficiency.

**KV Cache Eviction.** StreamingLLM (Xiao et al., 2023b) retains a few initial tokens along with recent tokens based on the observation of attention sinks, which can leading to the loss of critical information carried by the dropped tokens. H2O (Zhang et al., 2024c) selectively retains a small subset of tokens by greedily dropping those with lower contributions to cumulative attention. SnapKV (Li et al., 2024) selects clustered important KV positions for each attention head from an 'observation' window located at the end of the prompts. FastGen (Ge et al., 2023) adaptively evicts tokens from attention heads that focus on local contexts, discarding non-special tokens that surround key tokens, while standard KV cache is applied to attention heads that attend more broadly. PyramidKV (Zhang et al., 2024b) and PyramidInfer (Yang et al., 2024) take a hierarchical approach, adjusting KV cache sizes across different layers by allocating more cache to lower layers and less to higher ones.

**KV Cache Quantization.** SmoothQuant (Xiao et al., 2023a) enables the quantization of the KV cache to 8-bit with minimal performance degradation. Q-Hitter (Zhang et al., 2024c) leverages accumulated attention scores and "Quantization Friendliness" metrics to identify tokens that are crucial for preserving the generalization capabilities of LLMs, making them suitable for KV cache quantization. Furthermore, recent studies suggest that the key and value caches should be quantized differently (Liu et al., 2024b; Hooper et al., 2024): the key cache should be quantized per-channel, while the value cache should be quantized per-token.

**Structured Pruning of LLMs.** Traditional structured pruning (Ma et al., 2023; Ding et al., 2023; Lu et al., 2024a) of LLMs typically focuses on removing unimportant layers, heads, or hidden dimensions, often leading to significant performance degradation. In contrast, our approach preserves the original architecture of the LLM and specifically targets the channel dimension within each head's key cache. By dynamically identifying unimportant channels using data dependant criterion, our method greatly reduce the key cache size with minimal performance loss.

## 6    CONCLUSION AND LIMITATIONS

Motivated by the observation that certain channels exhibit significantly larger magnitudes and the low-rank nature of the key cache (as shown by singular value analysis), we propose THINK as a query-dependent pruning method targeting key cache channels. Optimized based on attention scores, our approach preserves essential information for each input query. THINK seamlessly integrates with existing token-level KV cache pruning (Li et al., 2024; Zhang et al., 2024c) and KV cache quantization (Liu et al., 2024b), further improving inference efficiency. Extensive experiments on LongBench and Needle-in-a-Haystack benchmarks demonstrate its effectiveness, achieving comparable or superior performance to baselines while reducing key cache size by 40%. However, our method slightly increases TTFT (Time-to-First-Token) due to channel importance computation (see Appendix F). Additionally, while effective for key cache pruning, value cache pruning remains an open direction for future work.

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

## A OBSERVATIONS

Figure 5 and Figure 4 illustrates the observations which motivates our approach THINK to prune unimportant key cache channels. We conducted singular value decomposition (SVD) (Demmel, 1997) on the attention weights of the specified layers to investigate their principal components. Note that

$$\mathbf{U}, \mathbf{\Sigma}, \mathbf{V} = \text{SVD}\left(\text{softmax}\left(\frac{\mathbf{Q}\mathbf{K}^T}{\sqrt{D}}\right)\right), \quad \text{Energy}_i = \frac{\sigma_i^2}{\sum_i \sigma_i^2}.$$

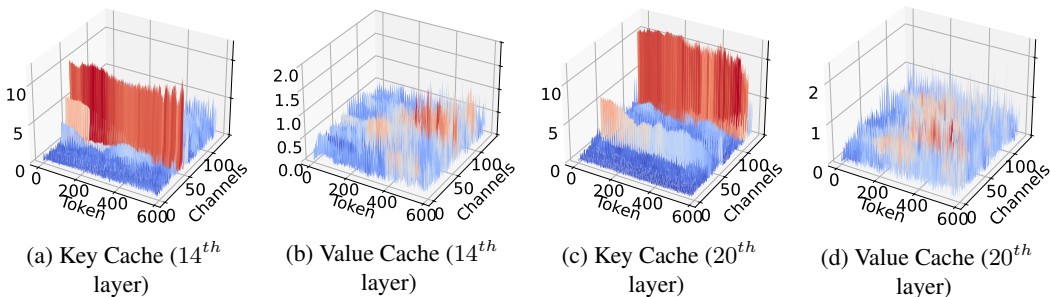

(a) Key Cache ($14^{th}$ layer)

(b) Value Cache ($14^{th}$ layer)

(c) Key Cache ($20^{th}$ layer)

(d) Value Cache ($20^{th}$ layer)

Figure 4: Magnitude of key and value cache for LLaMA-2-7B. The first head of layer 14 and layer 20 of LLaMA-2-7B is selected to visualize the magnitude of the key and value caches. We observe that the magnitudes of the key cache channels vary differently, whereas the channels of the value cache do not exhibit such variation.

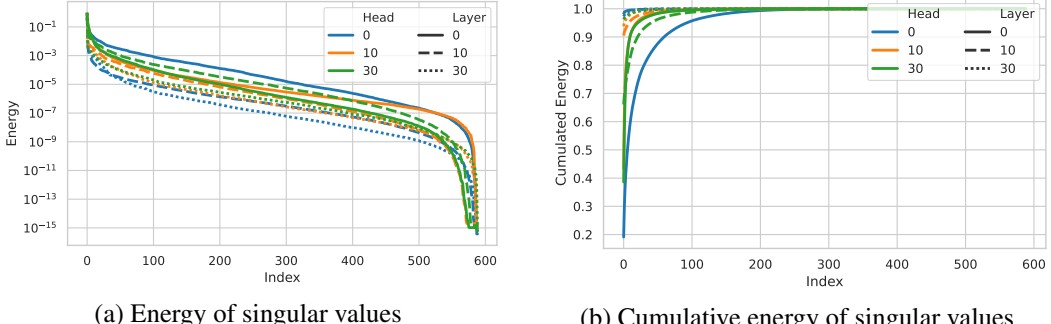

(a) Energy of singular values

(b) Cumulative energy of singular values

Figure 5: The energy and cumulative energy of the singular values.

# B NEEDLE-IN-A-HAYSTACK TEST PERFORMANCE COMPARISON

Figure 6 visualizes the test performance comparison on Needle-in-a-Haystack on Mistral-7B-Instruct-v0.2.

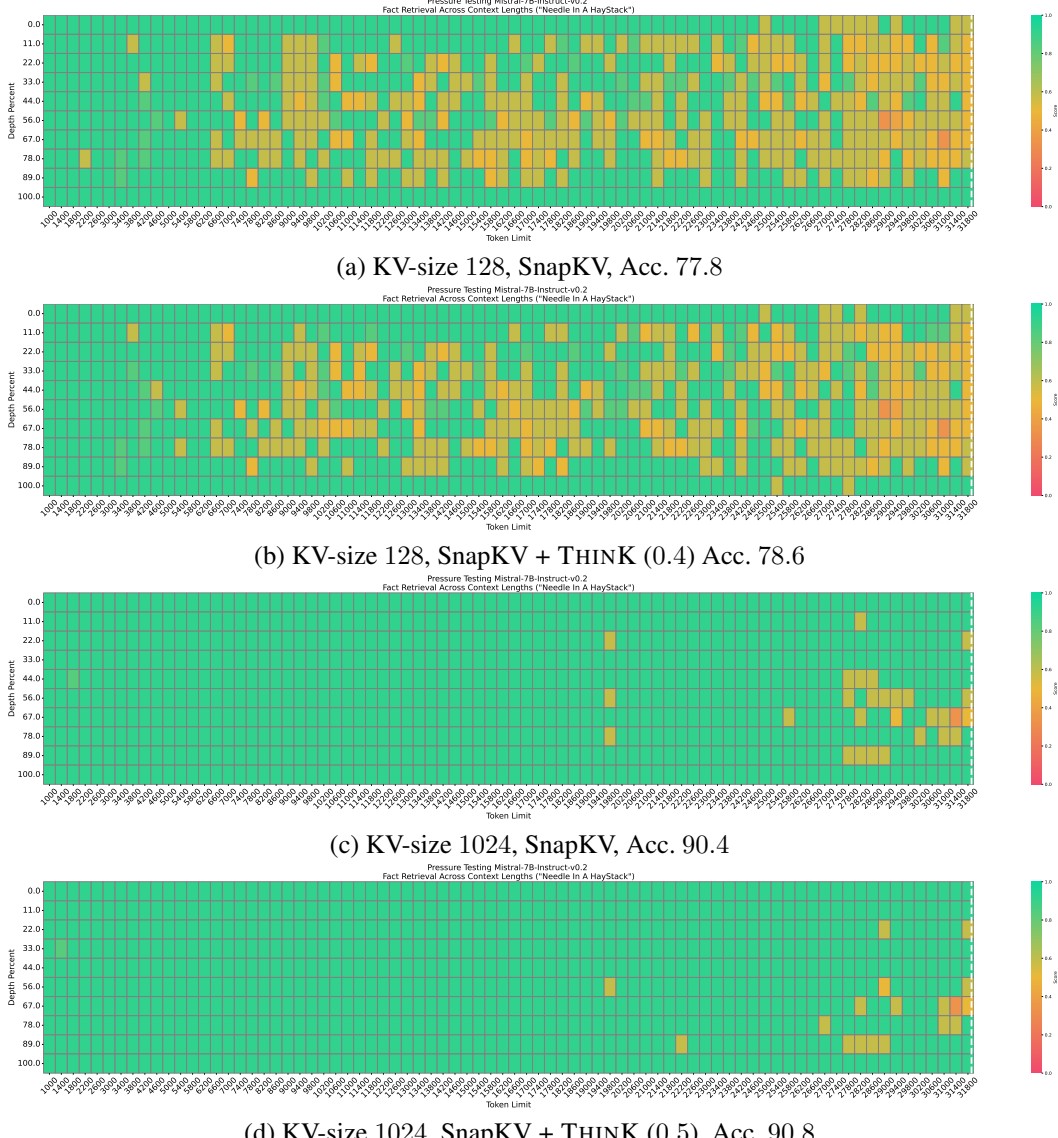

(a) KV-size 128, SnapKV, Acc. 77.8

(b) KV-size 128, SnapKV + THINK (0.4) Acc. 78.6

(c) KV-size 1024, SnapKV, Acc. 90.4

(d) KV-size 1024, SnapKV + THINK (0.5), Acc. 90.8

Figure 6: Needle-in-a-Haystack test performance comparison with Mistral-7B-Instruct-v0.2. THINK ($\lambda$) indicates we prune the key cache channels with a pruning ratio of $\lambda$

## C    IMPLEMENTATIONS

### C.1    IMPLEMENTATION WITH QUANTIZATION

Figure 7 illustrates the implementation of our method when integrated with the KV cache quantization method KIVI (Liu et al., 2024b). During the prefill phase, we first prune the unimportant channels of $X_K$ before applying quantization. In the decoding phase, each newly arrived key cache $t_K$ is added to $X_{K_r}$. Once $X_{K_r}$ reaches $G$ tokens, the residual length hyperparameter in KIVI, we prune and quantize the data, then concatenate it with the previously quantized $Q(P(X_{K_g}))$.

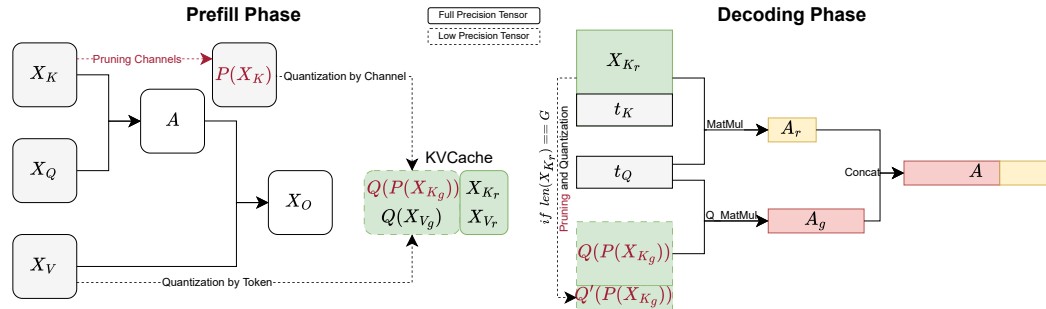

Figure 7: Implementations of THINK when incorporated with KIVI.

## D    VALUE CACHE PRUNING

Similar to the approach used for the Key cache, the pruning of channels in the Value cache can be guided by two primary criteria: magnitude-based pruning and query-driven pruning. We find that query-driven pruning is still better than magnitude based pruning.

$$\text{Score}_{v,i}(\mathbf{Q}_i, \mathbf{K}_i, \mathbf{V}_i)[j] = \|\text{softmax}(\frac{\mathbf{Q}_i[-S_{\text{obs}} :]\mathbf{K}_i^T}{\sqrt{D}})\mathbf{V}_i[:, j]\|_F \tag{3}$$

$$I_i = \mathbf{Top}_T(\text{Score}_{v,i}, T) \tag{4}$$

where $\mathbf{Q}_i, \mathbf{K}_i, \mathbf{V}_i \in \mathbb{R}^{S \times D}$. We define a criterion $\text{Score}_{v,i}$ to indicate the importance of each channel in the head $i$ of value cache. Then, only top T channels are retained. Table 8 reported the results of pruning both key and value channels, showing that pruning the Value cache channels is harder than pruning the Key cache channels. However, pruning 30% of both the Key cache and Value cache on LLaMA-3-8B-Instruct still results in acceptable performance. This demonstrates that the Value cache also has potential for pruning in the channel dimension. As depicted in Figure 4, the magnitude of the Key cache along the channel dimension is highly unbalanced, whereas the magnitude of the Value cache along the channel dimension is more uniform. This indicates that the channel sparsity in the Value cache is not as significant as in the Key cache, making it more challenging to identify redundant channels for pruning. We will investigate the value cache pruning strategy as part of our future work.

## E    PRUNING KEY CACHE ON VANILLA MODELS

To furthur demonstrate the effectiveness of our proposed THINK, we conducted additional experiments applying THINK directly to vanilla models, specifically LLaMA-3-8B-Instruct and Mistral-7B-Instruct-v0.2. The results, as shown in Table 9, demonstrate that our method maintains superior performance even after pruning 40% of the key cache channels. Furthermore, when the pruning ratio is increased to 50%, the performance degradation remains within an acceptable range. These findings further validate the effectiveness of THINK, not only for pruned models but also when applied directly to vanilla models.

Table 8: Performance comparison of pruning both K and V cache with different pruning ratios on LongBench. H2O + THINKV ($\lambda_1+\lambda_2$) indicates that the key cache channels of H2O are pruned with a pruning ratio of $\lambda_1$ and the value cache channels are pruned of a pruning ratio of $\lambda_2$.

| Method | Single-Document QA | | | Multi-Document QA | | | Summarization | | | Few-shot Learning | | | Synthetic | | Code | | Avg. |
|---|---|---|---|---|---|---|---|---|---|---|---|---|---|---|---|---|---|
| | NrtvQA | Qasper | MF-en | HotpotQA | 2WikiMQA | Musique | GovReport | QMSum | MultiNews | TREC | TriviaQA | SAMSum | PCount | PRe | Lcc | RB-P | |
| **LLaMA-3-8B-Instruct** | | | | | | | | | | | | | | | | | |
| KV-size 128 | | | | | | | | | | | | | | | | | |
| H2O | 22.12 | 13.20 | 31.61 | 37.79 | 32.71 | 18.45 | 20.32 | 22.02 | 21.10 | 38.50 | 87.75 | 39.14 | 5.83 | 69.50 | 55.06 | 50.97 | 35.38 |
| +THINKV (0.3+0.3) | 23.71 | 13.65 | 33.08 | 41.86 | 29.88 | 18.04 | 19.60 | 21.65 | 20.26 | 38.00 | 86.08 | 38.61 | 5.16 | 69.50 | 57.59 | 55.19 | **35.74** |
| SnapKV | 21.19 | 13.55 | 32.64 | 38.75 | 29.64 | 18.73 | 18.98 | 21.62 | 20.26 | 45.00 | 88.36 | 37.64 | 5.13 | 68.85 | 55.84 | 51.82 | 35.50 |
| +THINK(0.3+0.3) | 21.86 | 13.79 | 33.26 | 40.93 | 29.39 | 19.22 | 18.81 | 21.30 | 19.26 | 41.50 | 87.00 | 37.95 | 5.78 | 69.50 | 57.84 | 55.62 | **35.81** |
| KV-size 512 | | | | | | | | | | | | | | | | | |
| H2O | 23.52 | 17.93 | 34.68 | 42.11 | 33.52 | 19.92 | 22.11 | 22.56 | 23.82 | 41.00 | 90.46 | 40.20 | 5.87 | 69.50 | 56.71 | 51.69 | **37.23** |
| +THINKV (0.3+0.3) | 22.83 | 17.57 | 34.18 | 42.67 | 33.52 | 19.95 | 21.17 | 22.23 | 22.82 | 38.50 | 90.11 | 39.08 | 5.21 | 69.0 | 59.99 | 56.83 | **37.23** |
| SnapKV | 24.84 | 23.96 | 38.77 | 42.75 | 34.55 | 20.87 | 22.26 | 22.61 | 23.97 | 70.00 | 90.52 | 40.29 | 5.81 | 69.50 | 59.04 | 51.81 | 40.10 |
| +THINKV (0.3+0.3) | 24.57 | 24.59 | 38.09 | 44.61 | 34.37 | 20.37 | 21.23 | 21.95 | 23.30 | 66.00 | 90.69 | 39.38 | 5.60 | 69.00 | 61.75 | 58.46 | **40.25** |
| KV-size 2048 | | | | | | | | | | | | | | | | | |
| H2O | 25.56 | 26.85 | 39.54 | 44.30 | 32.92 | 21.09 | 24.68 | 23.01 | 26.16 | 53.00 | 90.65 | 41.84 | 4.91 | 69.25 | 58.43 | 51.31 | **39.59** |
| +THINKV (0.3+0.3) | 25.03 | 26.77 | 39.68 | 42.12 | 33.08 | 19.59 | 23.00 | 22.89 | 25.27 | 51.00 | 91.11 | 40.58 | 5.23 | 69.00 | 61.12 | 57.95 | **39.59** |
| +THINKV (0.4+0.4) | 24.87 | 24.31 | 37.77 | 43.13 | 34.42 | 19.60 | 21.67 | 22.70 | 24.52 | 49.00 | 90.81 | 39.28 | 6.00 | 69.00 | 61.81 | 58.08 | 39.19 |
| SnapKV | 25.86 | 29.55 | 41.10 | 44.99 | 35.80 | 21.81 | 25.98 | 23.40 | 26.46 | 73.50 | 90.56 | 41.66 | 5.17 | 69.25 | 58.67 | 51.52 | **41.58** |
| +THINKV (0.3+0.3) | 25.13 | 29.97 | 40.35 | 44.12 | 34.64 | 19.94 | 23.62 | 23.03 | 25.30 | 72.50 | 90.78 | 39.46 | 5.35 | 69.00 | 61.50 | 57.91 | 41.41 |
| +THINKV (0.4+0.4) | 25.13 | 28.85 | 40.70 | 44.21 | 36.36 | 21.07 | 22.31 | 22.89 | 24.80 | 72.50 | 90.88 | 38.77 | 6.41 | 69.00 | 61.49 | 58.87 | 41.52 |
| **Mistral-7B-Instruct-v0.2** | | | | | | | | | | | | | | | | | |
| KV-size 128 | | | | | | | | | | | | | | | | | |
| H2O | 21.21 | 21.81 | 38.87 | 30.42 | 20.36 | 12.30 | 20.58 | 22.61 | 22.10 | 39.00 | 82.37 | 40.44 | 2.90 | 79.56 | 51.22 | 48.38 | **34.63** |
| +THINKV (0.3+0.3) | 20.71 | 21.49 | 38.01 | 30.66 | 22.28 | 13.87 | 20.13 | 22.45 | 21.07 | 38.50 | 82.20 | 38.69 | 2.94 | 78.56 | 51.55 | 48.28 | 34.46 |
| SnapKV | 19.17 | 21.40 | 42.93 | 36.76 | 22.44 | 15.86 | 19.16 | 21.84 | 21.55 | 47.50 | 84.15 | 40.24 | 2.30 | 68.26 | 52.31 | 48.80 | **35.29** |
| +THINKV (0.3+0.3) | 19.92 | 20.61 | 42.68 | 37.63 | 23.19 | 15.09 | 18.97 | 21.93 | 20.55 | 45.00 | 84.06 | 39.33 | 2.99 | 66.00 | 51.51 | 47.51 | 34.81 |
| KV-size 512 | | | | | | | | | | | | | | | | | |
| H2O | 21.83 | 26.00 | 44.69 | 32.46 | 23.05 | 14.69 | 23.53 | 23.06 | 24.59 | 42.00 | 85.22 | 41.49 | 3.40 | 86.20 | 54.78 | 51.09 | **37.38** |
| +THINKV (0.3+0.3) | 22.36 | 24.26 | 44.77 | 30.47 | 22.94 | 14.96 | 22.63 | 22.90 | 23.73 | 41.50 | 85.30 | 40.21 | 3.08 | 80.07 | 54.48 | 50.96 | 36.54 |
| +THINKV (0.3+0.1) | 22.14 | 25.15 | 45.29 | 31.78 | 23.21 | 14.62 | 23.36 | 22.70 | 24.51 | 41.50 | 85.61 | 41.58 | 2.75 | 84.03 | 54.50 | 51.09 | 37.11 |
| SnapKV | 24.44 | 27.81 | 48.98 | 39.46 | 25.25 | 16.98 | 23.70 | 22.76 | 24.37 | 71.26 | 85.88 | 41.26 | 2.78 | 86.56 | 56.46 | 53.41 | **40.46** |
| +THINKV (0.3+0.3) | 24.10 | 27.04 | 47.76 | 38.66 | 25.45 | 17.51 | 22.64 | 22.81 | 23.91 | 66.00 | 86.62 | 39.91 | 3.36 | 82.24 | 55.96 | 52.81 | 39.80 |
| +THINKV (0.3+0.1) | 23.90 | 28.14 | 48.35 | 39.03 | 24.83 | 16.68 | 23.51 | 23.12 | 24.34 | 67.50 | 86.09 | 41.69 | 2.65 | 84.34 | 57.29 | 53.22 | 40.29 |
| KV-size 1024 | | | | | | | | | | | | | | | | | |
| H2O | 23.67 | 28.55 | 46.4 | 36.99 | 24.82 | 15.02 | 25.21 | 23.04 | 25.77 | 46.00 | 85.93 | 41.98 | 3.24 | 86.57 | 56.40 | 52.75 | **38.90** |
| +THINKV (0.3+0.3) | 23.65 | 26.54 | 47.00 | 35.52 | 24.79 | 17.15 | 23.64 | 23.12 | 25.20 | 44.00 | 86.38 | 41.67 | 3.46 | 80.14 | 56.53 | 52.86 | 38.23 |
| +THINKV (0.3+0.1) | 24.13 | 28.57 | 46.31 | 35.59 | 24.92 | 15.34 | 24.58 | 23.33 | 25.93 | 45.50 | 85.91 | 42.97 | 2.57 | 83.64 | 55.39 | 52.73 | 38.59 |
| SnapKV | 25.47 | 29.57 | 49.33 | 40.90 | 25.53 | 19.01 | 25.94 | 23.89 | 26.21 | 69.50 | 86.48 | 42.10 | 2.98 | 88.56 | 57.19 | 53.60 | **41.64** |
| +THINKV (0.3+0.3) | 25.29 | 29.25 | 49.56 | 41.25 | 25.75 | 19.37 | 24.64 | 23.02 | 25.27 | 69.00 | 86.70 | 40.92 | 3.29 | 82.06 | 57.15 | 54.15 | 41.02 |
| +THINKV (0.3+0.1) | 25.84 | 29.30 | 49.56 | 41.44 | 25.29 | 19.02 | 25.21 | 23.73 | 25.72 | 69.00 | 86.69 | 42.55 | 2.44 | 85.76 | 57.55 | 54.10 | 41.45 |
| KV-size 2048 | | | | | | | | | | | | | | | | | |
| H2O | 25.76 | 31.10 | 49.06 | 40.38 | 26.43 | 16.78 | 27.17 | 23.64 | 26.69 | 55.0 | 86.35 | 42.48 | 2.72 | 86.64 | 56.98 | 53.91 | **40.69** |
| +THINKV (0.3+0.3) | 25.60 | 28.74 | 47.54 | 38.67 | 26.25 | 17.35 | 24.54 | 23.27 | 26.15 | 51.00 | 87.01 | 43.02 | 2.94 | 81.46 | 56.41 | 54.26 | 39.64 |
| +THINKV (0.3+0.1) | 25.64 | 30.65 | 48.95 | 40.42 | 26.43 | 16.65 | 26.76 | 23.51 | 26.59 | 52.50 | 86.53 | 43.45 | 2.66 | 83.96 | 56.55 | 53.83 | 40.32 |
| SnapKV | 25.89 | 32.56 | 48.55 | 41.68 | 27.24 | 18.75 | 28.90 | 24.47 | 26.63 | 70.00 | 86.27 | 42.57 | 3.09 | 86.93 | 57.44 | 53.83 | **42.18** |
| +THINKV (0.3+0.3) | 27.01 | 30.72 | 48.81 | 41.15 | 26.93 | 18.93 | 25.81 | 23.59 | 26.42 | 70.00 | 86.82 | 41.91 | 3.05 | 82.65 | 57.01 | 54.25 | 41.57 |
| +THINKV (0.3+0.1) | 26.22 | 32.69 | 48.96 | 40.83 | 26.70 | 19.02 | 27.87 | 24.23 | 26.64 | 70.00 | 86.65 | 42.63 | 2.22 | 85.13 | 57.00 | 54.28 | 41.94 |

Table 9: Performance comparison of pruning key cache on vanilla models on LongBench.

| Method | Single-Document QA | | | Multi-Document QA | | | Summarization | | | Few-shot Learning | | | Synthetic | | Code | | Avg. |
|---|---|---|---|---|---|---|---|---|---|---|---|---|---|---|---|---|---|
| | NrtvQA | Qasper | MF-en | HotpotQA | 2WikiMQA | Musique | GovReport | QMSum | MultiNews | TREC | TriviaQA | SAMSum | PCount | PRe | Lcc | RB-P | |
| LLaMA-3-8B-Instruct, KV-size Full | | | | | | | | | | | | | | | | | |
| Vanilla | 25.56 | 32.27 | 39.71 | 43.56 | 35.09 | 21.18 | 28.71 | 23.26 | 26.64 | 73.50 | 90.48 | 42.33 | 4.80 | 69.25 | 59.29 | 54.05 | 41.86 |
| +THINK(0.4) | 25.32 | 32.26 | 39.81 | 44.19 | 34.77 | 21.10 | 28.63 | 23.13 | 26.38 | 73.50 | 90.58 | 41.69 | 5.21 | 69.50 | 61.94 | 58.37 | **42.27** |
| +THINK(0.5) | 25.35 | 32.80 | 41.64 | 43.99 | 31.81 | 21.58 | 28.35 | 23.31 | 26.80 | 73.50 | 90.37 | 40.81 | 5.67 | 69.17 | 61.90 | 59.00 | 42.25 |
| +THINK(0.6) | 24.39 | 31.16 | 41.80 | 42.52 | 31.63 | 20.70 | 25.84 | 23.18 | 25.46 | 73.50 | 90.43 | 38.97 | 5.77 | 68.46 | 59.63 | 59.38 | 41.43 |
| Mistral-7B-Instruct-v0.2, KV-size Full | | | | | | | | | | | | | | | | | |
| Vanilla | 26.63 | 32.99 | 49.34 | 42.77 | 27.35 | 18.77 | 32.87 | 24.24 | 27.10 | 71.00 | 86.23 | 42.96 | 2.75 | 86.98 | 56.93 | 54.49 | 42.71 |
| +THINK(0.4) | 27.11 | 33.46 | 48.73 | 41.79 | 28.14 | 18.87 | 32.45 | 24.55 | 27.09 | 71.00 | 86.26 | 43.02 | 3.95 | 86.31 | 56.99 | 54.36 | **42.76** |
| +THINK(0.5) | 26.63 | 33.71 | 49.38 | 42.38 | 26.78 | 18.76 | 32.57 | 24.63 | 26.92 | 71.00 | 86.39 | 42.82 | 3.13 | 84.65 | 56.75 | 54.04 | 42.53 |
| +THINK(0.6) | 27.03 | 33.23 | 49.49 | 42.65 | 26.43 | 18.14 | 31.74 | 24.75 | 26.57 | 71.00 | 86.28 | 41.40 | 3.50 | 83.11 | 55.93 | 53.37 | 42.16 |

# F COMPARISONS OF GENERATION SPEED AND THROUGHPUT

In this section, we provide comparisons of TTFT (Time To First Token), TPOT (Time Per Output Token), memory usage, and throughput in Table 10 and Table 11. We follow the methodology used in KIVI Liu et al. (2024b). We generate synthetic workloads with an input prompt length of 160 and an output length of 338. We set a batch size 300 for both KIVI and our method. As our method performs online pruning, there is a slight delay introduced during the prefilling stage due to the computation of channel importance. However, the reduction in memory usage leads to notable improvements in TPOT during inference. Specifically, after applying THINK, TPOT improves from 0.27 ms/token to 0.25 ms/token with 40% pruning, and further improves to 0.24 ms/token with a

Table 10: Comparisons of TTFT (Time To First Token), TPOT (Time Per Output Token) and memory usage on LLaMA-2-7B.

| Method | KIVI(4/4) | KIVI(4/4) + THINK(0.4) | KIVI (4/4) + THINK(0.5) |
|---|---|---|---|
| **Memory (GB)** | 61.7 | 53.3 | 51.2 |
| **TTFT (ms)** | 7.0 | 10.4 | 10.2 |
| **TPOT (ms/token)** | 0.27 | 0.25 | 0.24 |

Table 11: Comparisons of throughput on LLaMA-2-7B.

| Method | Vanilla | KIVI(4/4) | KIVI(4/4) + THINK(0.4) | KIVI (4/4) + THINK(0.5) |
|---|---|---|---|---|
| **Throughput (tokens/s)** | 2557 | 5168 | 5518 | 5676 |

50% pruning ratio. Regarding throughput, with the same memory usage, THINK (40% pruning) improves the throughput of KIVI from 5168 tokens/s to 5518 tokens/s. At a pruning ratio of 50%, throughput increases further to 5676 tokens/s. Model weights and KV cache are the primary memory components accessed during generation. By effectively reducing the memory footprint of the KV cache, our method alleviates the memory bottleneck, enabling larger batch sizes and faster generation speeds. These results highlight the potential of THINK to enhance inference performance by balancing memory efficiency with computational overhead. We are continuing to optimize the implementation to further improve performance. Memory bandwidth is a major performance bottleneck in the decoding phase of large language models (LLMs).

## G    COMPARISONS WITH SVD BASED METHODS

In this section, we compare our THINK with SVD based methods (Saxena et al., 2024; Yuan et al., 2023; Chang et al., 2024). SVD based KV cache compression method decompose the KV cache weights offline with some calibration data, relying on storing latent representations and recovering the cache during inference. Our approach is an online pruning strategy that is plug-and-play, requiring no changes to the model's weights or architecture. This makes THINK lightweight and easy to integrate into existing systems. Besides, our method dynamically prunes the Key cache channels directly without requiring any reconstruction. We provide comparisons of our method with SVD-based approaches in Table 12. Following Palu, we use Mistral-7B-Instruct-v0.2 as the baseline model. Our results show that when the KV cache is compressed by 80%, our method preserves performance, whereas Palu experiences an accuracy drop of 1.28%. Similarly, for ASVD, we evaluate under the same compression rate. Our method demonstrates significantly better performance, with only a 0.12% performance drop compared to the 5.21% drop observed for ASVD. The results demonstrate the effectiveness of THINK, with significantly less performance degradation compared to SVD-based methods under the same compression rates. Our pruning method has the potential to be integrated with SVD-based approaches. Exploring such a combination could yield further advancements in KV cache compression, which we aim to investigate in future work.

Table 12: Performance comparison of SVD based methods on LongBench.

| Method | Single-Document QA | | | Multi-Document QA | | | Summarization | | | Few-shot Learning | | | Synthetic | | Code | | Avg. |
|---|---|---|---|---|---|---|---|---|---|---|---|---|---|---|---|---|---|
| | NrtvQA | Qasper | MF-en | HotpotQA | 2WikiMQA | Musique | GovReport | QMSum | MultiNews | TREC | TriviaQA | SAMSum | PCount | PRe | Lcc | RB-P | |
| | | | | | | | | Mistral-7B-Instruct-v0.2 | | | | | | | | | |
| Vanilla | 26.63 | 32.99 | 49.34 | 42.77 | 27.35 | 18.77 | 32.87 | 24.24 | 27.10 | 71.00 | 86.23 | 42.96 | 2.75 | 86.98 | 56.93 | 54.49 | 42.71 |
| +Palu | 27.47 | 34.11 | 48.47 | 44.09 | 26.33 | 20.18 | 31.15 | 24.30 | 27.12 | 70.00 | 85.80 | 41.74 | 2.74 | 73.18 | 51.70 | 54.52 | 41.43 |
| +THINK(0.4) | 27.11 | 33.46 | 48.73 | 41.79 | 28.14 | 18.87 | 32.45 | 24.55 | 27.09 | 71.00 | 86.26 | 43.02 | 3.95 | 86.31 | 56.99 | 54.36 | 42.76 |
| | | | | | | | | LLaMA-2-7B-Chat | | | | | | | | | |
| Vanilla | 18.39 | 20.11 | 35.67 | 31.25 | 25.50 | 10.14 | 25.68 | 20.93 | 26.27 | 64.00 | 83.38 | 40.99 | 5.50 | 10.00 | 60.81 | 55.27 | 33.37 |
| +ASVD | 16.46 | 13.19 | 28.98 | 21.94 | 22.86 | 8.74 | 17.73 | 20.27 | 22.35 | 57.00 | 73.88 | 38.51 | 1.50 | 4.77 | 53.32 | 49.13 | 28.16 |
| +THINK(0.4) | 18.39 | 19.98 | 35.05 | 30.85 | 25.65 | 10.25 | 25.98 | 20.82 | 26.04 | 64.00 | 83.63 | 41.55 | 6.00 | 8.50 | 60.18 | 55.18 | 33.25 |

