# OpenReview forum: "ThinK: Thinner Key Cache by Query-Driven Pruning"
_ICLR.cc/2025/Conference — ICLR 2025 Spotlight_

### Official Review · Reviewer_1rme · 2024-10-24

**Soundness:** 3
**Presentation:** 3
**Contribution:** 3
**Rating:** 8
**Confidence:** 5

**Summary:**

This paper propose a new way to prune KV cache that focuses on reducing least significant channels. The method is query-dependent and reduces KV cache size upon previous KV compression methods while maintaining almost the same quality with previous methods. It also discovers the fact that activated key cache is sparse for a given query.

**Strengths:**

1. Very extensive benchmark scenario. Evaluated with models from 7B to 70B on 10+ datasets. The accuracy drop is indeed negligible.
2. The method proposed is simple and works on top of previous methods. This could be a nice addition to memory-efficient transformer inference.
3. Writing is easy to follow and method description is clear.

**Weaknesses:**

1. Method is dependent on query but did not show online pruning delay costs. It would be nice to show a few numbers on long the pruning takes.
2. Did not discuss enough for value cache. This is understandable since value cache is indeed harder to compress according to my own experience. However, you should add more discussion on the implication with this. For example, you claimed that you have 2.8 peak memory saving, does this include saving with V cache? More discussion is needed on this thread to make readers better understand the use scenario of your method (this has nothing to do with the validity of your key cache pruning, it is just unclear to me how much actual benefit comes with your method and I am worried of readers may over-estimate your gain).
3. KIVI is not your contribution. When you present the result in abstraction, please report clearly your improvement SEPARATELY. What is your gain above KIVI? Compressing K cache by 60% without accuracy drop is a very good contribution already!

**Questions:**

1. Could you address the questions in weakness?
2. I think you should be more modest about your improvement. Your method is solid and could be impactful (ex. you potentially could also help reduce computation needed in attention calculation by reducing K cache channel dimension). I suggest to make it clear about what part is your contribution and what part is the previous methods (ex. in abstract), and dig deeper into why your improvement in this paper could benefit inference system-level performance.

---

> ### Author Response · Authors · 2024-11-21
> **Response to Reviewer 1rme (1/2)**
>
> >**Q1.** Method is dependent on query but did not show online pruning delay costs. It would be nice to show a few numbers on long the pruning takes.
>
> **A1.** Thank you for the suggestion! You are correct that our method involves online pruning, which incurs some delay cost in TTFT (Time To First Token). However, it is important to note that while there is a slight delay during the prefilling stage, the generation speed (measured by TPOT: Time Per Output Token) is significantly improved. We are actively working on optimizing the implementation to further minimize this delay.
> As shown in the following table, TPOT improves from 0.27 ms/token to 0.25 ms/token, and further improves to 0.24 ms/token when the pruning ratio is increased to 50%. Additionally, please note that all these experiments involve pruning only the Key cache channels, which demonstrates the benefits of our approach despite the minor TTFT delay. We will update the manuscript with these results in the revision for greater clarity.
>
> |**Methods**|**KIVI(4/4)**|**KIVI(4/4)+ThinK(0.4)**|**KIVI(4/4)+ThinK(0.5)**|
> |--------------|--------------|--------------|--------------|
> |**TTFT(ms)**| 7 | 10.4|10.2|
> |**TPOT(ms/token)**|0.27|0.25|0.24|
> |**Mem(GB)**|61.7|53.3|51.2|
>
> Table. Comparisons of TTFT (Time To First Token) and TPOT (Time Per Output Token) on LLaMA-2-7B-Chat.
>
> >**Q2.** Did not discuss enough for value cache. This is understandable since value cache is indeed harder to compress according to my own experience. However, you should add more discussion on the implication with this. For example, you claimed that you have 2.8 peak memory saving, does this include saving with V cache? More discussion is needed on this thread to make readers better understand the use scenario of your method (this has nothing to do with the validity of your key cache pruning, it is just unclear to me how much actual benefit comes with your method and I am worried of readers may over-estimate your gain).
>
> **A2** Thank you for pointing this out! We apologize for the lack of clarity. To clarify, our method focuses on pruning Key cache channels by default. All results presented in the main paper, including the reported memory reduction and peak memory savings, are based solely on Key cache pruning. Some discussions in Section 4.4 and results in Table 7 include additional experiments on pruning Value cache channels. In our paper, the method pruning Key cache alone is denoted as ThinK, while the method pruning both Key and Value cache is denoted as ThinKV.
>
> Based on our observations (e.g., unbalanced magnitude in the Key cache and low-rank properties in attention weights), we identified redundancy in the channel dimension of the Key cache. This motivated us to propose a query-driven pruning criterion specifically for Key cache channel pruning, which achieves significant memory savings without degrading performance. While we also conducted experiments on pruning the Value cache channels, we found that the performance impact was inconsistent across different models. For these reasons, Value cache pruning experiments are included in the appendix as exploratory results. We acknowledge the importance of further exploring how to better combine Key cache and Value cache pruning and will investigate this integration as part of our future work. We will include this discusion in the revised manuscript to avoid any potential overestimation of the gains from our method. **We will add notation in the beginning of Section 4 for greater clarity.**

---

> ### Author Response · Authors · 2024-11-21
> **Response to Reviewer 1rme (2/2)**
>
> >**Q3** KIVI is not your contribution. When you present the result in abstraction, please report clearly your improvement SEPARATELY. What is your gain above KIVI? Compressing K cache by 60% without accuracy drop is a very good contribution already!
>
> **A3.** Thank you for recognizing our contribution and for the helpful suggestion. We have clarified the improvements brought by ThinK when integrated with KIVI:
> * Batch Size Improvement: After integrating ThinK, the achievable batch size increases from 4$\times$ (with KIVI alone) to 5$\times$, demonstrating a notable improvement in memory efficiency.
> * Peak Memory Reduction: ThinK reduces the peak memory usage from 61.7 GB (KIVI alone) to 53.3 GB, further enhancing the practicality of KV cache compression in resource-constrained environments.
>
> We will ensure that these improvements are clearly reported separately from KIVI in the revised manuscript to avoid ambiguity and properly highlight the contributions of ThinK.
>
>
> >**Q4.** I think you should be more modest about your improvement. Your method is solid and could be impactful (ex. you potentially could also help reduce computation needed in attention calculation by reducing K cache channel dimension). I suggest to make it clear about what part is your contribution and what part is the previous methods (ex. in abstract), and dig deeper into why your improvement in this paper could benefit inference system-level performance.
>
> **A4.** Thank you for your valuable feedback and for recognizing the potential of our method. We apologize if the description of our contributions was unclear. **In response, we have revised the descriptions in Abstract, Ablation Studies, and Conclusion to explicitly delineate our contributions from previous methods.** We will also expand the discussion in the revised manuscript to highlight how our improvements can benefit inference system-level performance:
>
> Memory bandwidth is a major performance bottleneck in the decoding phase of large language models (LLMs). Model weights and KV cache are the primary memory components accessed during generation. By effectively reducing the memory footprint of the KV cache, our method alleviates the memory bottleneck, enabling larger batch sizes and faster generation speeds. As shown in the table provided in A1, our method improves the generation speed (measured by TPOT: Time Per Output Token) by reducing the KV cache size. This demonstrates the practical benefits of our approach in optimizing inference throughput. Our method introduces a novel optimization dimension for KV cache—channel pruning. This opens new avenues for system-level improvements in memory efficiency and computational performance that can be further explored in future work.
>
> We appreciate the opportunity to improve our presentation and will ensure these points are clearly articulated in the revised manuscript.

---

> > ### Comment · Reviewer_1rme · 2024-11-21
> >
> > Thank you for the clarification and the experiments added. I hope you could add the TTFT increase in your paper if it gets accepted in the end. I think your work is solid and indeed give a new tradeoff in the existing space. The improvement is not extremely significant but it could be a nice add on. I am changing it to an accept but I hope you make it clear about the limitations (TTFT increase, only Key cache memory reduction, xxx)

---

> > > ### Author Response · Authors · 2024-11-22
> > >
> > > Thank you for acknowledging our response and efforts! We will include the TTFT increase in the final version and clearly outline the limitations of our method.

---

### Official Review · Reviewer_ssq7 · 2024-11-03

**Soundness:** 3
**Presentation:** 3
**Contribution:** 2
**Rating:** 5
**Confidence:** 5

**Summary:**

This paper proposes ThinK, a novel KV cache compression technique that uses query-driven channel pruning for keys and values. Specifically, channel importance is established through $|| Q.K^T ||$ magnitude, and channels with the highest scores are selected. Extensive experiments over the Llama and Mistral family of models demonstrate a 20% reduction in KV cache size when ThinK is used in conjunction with existing KV cache compression techniques.

**Strengths:**

1. The paper is well-written and concise, which makes it relatively easy to follow.
2. The authors conduct an extensive set of experiments (along with ablation studies) to show performance improvements in the form of reduced KV cache footprint as well as the possibility of enabling larger batch sizes through their technique.

**Weaknesses:**

1. Authors overlook relevant baselines [1-3] which also reduce KV cache size by focusing on the hidden dimension (hence effectively pruning channels). Comparison with these techniques is important to put this work in context with other such KV cache compression techniques.

2. The performance benefits are not very significant since the authors show key cache compression by up to 40% and value cache up to 30%, leading to only ~20% improvement in KV cache memory footprint.

[1] Saxena et al., "Eigen Attention: Attention in Low-Rank Space for KV Cache Compression.", ArXiv 2024.

[2] Yuan et al., "ASVD: Activation-aware Singular Value Decomposition for Compressing Large Language Models", ArXiv 2024.

[3] Chang et al., "Palu: Compressing KV-Cache with Low-Rank Projection", ArXiv 2024.

**Questions:**

1. Which dataset was used to find channel importance as shown in Section 3.2? Is a pretaining dataset like WikiText a good fit for establishing channel importance for a range of downstream tasks, or task-specific data is necessary? It would be helpful to understand the generalizability of channel importance score S.

2. Is query-driven pruning a better metric than SVD (as used in Eigen Attention [1]) to find channel importance? Comparison with such works can further clarify the superiority and/or limitations of ThinK.

3. (Suggestion) A pruning ratio ($\lambda$) of 0.4 implies a 40% reduction in the key cache, which means a 20% reduction for the KV cache. This should be specified in Section 4.2.

4. Can the authors provide some discussion on why it's harder to compress the value cache as compared to the key cache (Table 1-7 vs Table 8 in Appendix)?

---

> ### Author Response · Authors · 2024-11-21
> **Response to Reviewer ssq7 (1/2)**
>
> We thank the reviewer for the comments and questions.
>
> >**Q1.** Authors overlook relevant baselines [1-3] which also reduce KV cache size by focusing on the hidden dimension (hence effectively pruning channels). Comparison with these techniques is important to put this work in context with other such KV cache compression techniques.
>
> **A1.** Thank you for the suggestion. It is worth noting that ASVD[2] and Palu[3] are also ICLR submissions: [ASVD](https://openreview.net/forum?id=HyPofygOCT&noteId=vwBXfcYxgN) and [Palu](https://openreview.net/forum?id=LWMS4pk2vK), but we have still conducted comparisons with ASVD and Palu on LongBench in the additional tables provided. Following Palu, we use Mistral-7B-Instruct-v0.2 as the baseline model. Our results show that when the KV cache is compressed by 80%, our method preserves performance, whereas Palu experiences an accuracy drop of 1.28%. Similarly, for ASVD, we evaluate under the same compression rate. Our method demonstrates significantly better performance, with only a 0.12% performance drop compared to the 5.21% drop observed for ASVD. A key distinction is that these baselines reduce KV cache size by applying SVD to the weight matrix offline. SVD-based methods store internal latent representations and require recovering the cache during inference. In contrast, our approach operates online, dynamically pruning Key cache channels without changing the structure of models or requiring reconstruction during inference. Additionally, our method has the potential to integrate with SVD-based approaches, enabling further reduction of decomposed dimensions and enhanced compression efficiency.
>
> | **Method**|Compression Rate|**Avg.** |**NrtvQA** | **Qasper** | **MF-en** | **HotpotQA** | **2WikiMQA** | **Musique** | **GovReport** | **QMSum** | **MultiNews** | **TREC** | **TriviaQA** | **SAMSum** | **PCount** | **PRe** | **Lcc** | **RB-P** |
> |--------------|----|-----|-----|------------|-----------------|---------------|-------------|---------------|-----------|---------------|----------|--------------|-------------|------------|---------|----------|-----------|-------|
> |Vanilla |-|42.71|26.63 | 32.99 | 49.34 | 42.77 | 27.35 | 18.77 | 32.87 | 24.24 | 27.10 | 71.00 | 86.23 | 42.96 | 2.75 | 86.98 | 56.93 | 54.49 |
> |Vanilla+THINK(0.4)|0.8|42.76|27.11 | 33.46 | 48.73 | 41.79 | 28.14 | 18.87 | 32.45 | 24.55 | 27.09 | 71.00 | 86.26 | 43.02 | 3.95 | 86.31 | 56.99 | 54.36 |
> |Vanilla+Palu[3]|0.8|41.43|27.47 | 34.11 | 48.47 | 44.09 | 26.33 | 20.18 | 31.15 | 24.30 | 27.12 | 70.00 | 85.80 | 41.74 | 2.74 | 73.18 | 51.70 | 54.52 |
>
> Table. Performance comparison of ThinK and Palu on Mistral-7B-Instruct-v0.2 on LongBench.
>
> | **Method**|Compression Rate|**Avg.** |**NrtvQA** | **Qasper** | **MF-en** | **HotpotQA** | **2WikiMQA** | **Musique** | **GovReport** | **QMSum** | **MultiNews** | **TREC** | **TriviaQA** | **SAMSum** | **PCount** | **PRe** | **Lcc** | **RB-P** |
> |--------------|----|-----|-----|------------|-----------------|---------------|-------------|---------------|-----------|---------------|----------|--------------|-------------|------------|---------|----------|-----------|-------|
> |Vanilla |-|33.37|18.39 | 20.11 | 35.67 | 31.25 | 25.50 | 10.14 | 25.68 | 20.93 | 26.27 | 64.00 | 83.38 | 40.99 | 5.50 | 10.00 | 60.81 | 55.27 |
> |Vanilla+THINK(0.4)|0.8|33.25|18.39 | 19.98 | 35.05 | 30.85 | 25.65 | 10.25 | 25.98 | 20.82 | 26.04 | 64.00 | 83.63 | 41.55 | 6.00 | 8.50 | 60.18 | 55.18 |
> |Vanilla+ASVD[2]|0.8|28.16|16.46 | 13.19 | 28.98 | 21.94 | 22.86 | 8.74 | 17.73 | 20.27 | 22.35 | 57.00 | 73.88 | 38.51 | 1.50 | 4.77 | 53.32 | 49.13 |
>
> Table. Performance comparison of ThinK and ASVD on LLaMA-2-7B-Chat on LongBench.
>
> [1] Saxena et al., "Eigen Attention: Attention in Low-Rank Space for KV Cache Compression.", ArXiv 2024.
>
> [2] Yuan et al., "ASVD: Activation-aware Singular Value Decomposition for Compressing Large Language Models", ArXiv 2024.
>
> [3] Chang et al., "Palu: Compressing KV-Cache with Low-Rank Projection", ArXiv 2024.

---

> ### Author Response · Authors · 2024-11-21
> **Response to Reviewer ssq7 (2/2)**
>
> >**Q2.** The performance benefits are not very significant since the authors show key cache compression by up to 40% and value cache up to 30%, leading to only ~20% improvement in KV cache memory footprint.
>
> **A2.** Pruning 40% of the Key cache channels alone results in a ~20% reduction in KV cache memory. If 30% of the Value cache channels are further pruned, the total memory reduction increases to ~35%. While the main part of our paper primarily focuses on pruning the Key cache, we believe that even a 20% memory reduction is both remarkable and significant, as also recognized by Reviewer #1rme. Here’s why:
>
> * **Complementary Optimization**: KV cache optimization should be approached from multiple dimensions. In this work, we propose a novel optimization dimension—channel pruning—that can be integrated with existing methods such as KV eviction and KV quantization for additional benefits. For instance, the original KIVI method achieves a 4× batch size on a single GPU. With the addition of ThinK, this batch size is further increased to 5×, which is a substantial improvement.
> * **Future Potential**: Our work demonstrates the potential of optimizing the KV cache along the channel dimension, opening the door to further advancements in this area.
> * **Practical Impact**: The serving costs of large language models are immense, especially given the sheer volume of requests in production environments. A 20% reduction in memory usage translates to significant resource savings and efficiency improvements at scale.
>
> >**Q3.** Which dataset was used to find channel importance as shown in Section 3.2? Is a pretaining dataset like WikiText a good fit for establishing channel importance for a range of downstream tasks, or task-specific data is necessary?
>
> **A3.** Thanks for your question. Our method employs an **online pruning** strategy, meaning that we do not require calibration data or pretraining datasets to establish channel importance. The channel importance score $S$ is computed dynamically for each context input, ensuring that it adapts to the specific characteristics of the data being processed. This approach inherently generalizes well across different tasks, as the computation of $S$ is context-dependent and does not rely on pre-computed or static importance scores. We hope this clarifies the generalizability and flexibility of our method.
>
> >**Q4.** Is query-driven pruning a better metric than SVD (as used in Eigen Attention [1]) to find channel importance?
>
> **A4.** Thanks for the question. Query-driven pruning offers several advantages over SVD-based methods:
>
> * Our approach is an online pruning strategy that is plug-and-play, requiring no changes to the model's weights or architecture. This makes ThinK lightweight and easy to integrate into existing systems.
> * SVD-based methods rely on storing latent representations and recovering the cache during inference. In contrast, our method dynamically prunes the Key cache channels directly without requiring any reconstruction.
> * We have provided comparisons of our method with SVD-based approaches, such as those referenced in A1. The results demonstrate the effectiveness of ThinK, with significantly less performance degradation compared to SVD-based methods under the same compression rates.
> * Our pruning method has the potential to be integrated with SVD-based approaches. Exploring such a combination could yield further advancements in KV cache compression, which we aim to investigate in future work.
>
> We believe these points clarify the advantages of query-driven pruning and highlight its effectiveness in optimizing KV cache memory while maintaining model performance.
>
> >**Q5.** A pruning ratio ($\lambda$) of 0.4 implies a 40% reduction in the key cache, which means a 20% reduction for the KV cache. This should be specified in Section 4.2.
>
> **A5.** Thanks for the suggestion, we will update in the revision.
>
> >**Q6** Can the authors provide some discussion on why it's harder to compress the value cache?
>
> **A6.** Thanks for this suggestion. We have included the following discussions in the revision.
>
> As depicted in Figure 4, the magnitude of the Key cache along the channel dimension is highly unbalanced, whereas the magnitude of the Value cache along the channel dimension is more uniform. This indicates that the channel sparsity in the Value cache is not as significant as in the Key cache, making it more challenging to identify redundant channels for pruning. However, this does not mean that the Value cache cannot be pruned. As shown in Table 8, pruning 30% of both the Key cache and Value cache on LLaMA-3-8B-Instruct still results in acceptable performance. This demonstrates that the Value cache also has potential for pruning in the channel dimension. We acknowledge the importance of further exploring the integration of Key and Value cache pruning strategies and will investigate this as part of our future work.

---

> ### Author Response · Authors · 2024-11-25
> **Looking forward to your reply**
>
> Dear Reviewer ssq7,
>
> We hope this message finds you well. As the author-reviewer discussion period approaches, we respectfully seek your confirmation on the adequacy of our rebuttal in addressing the concerns raised in your review.
>
> We really appreciate the substantial time and effort you have committed to reviewing our work and are grateful for the additional insights. Your comments have been very helpful in refining our project.
>
> Thank you once again for your valuable perspectives. We eagerly await your further guidance.
>
> Sincerely,
>
> Authors

---

### Official Review · Reviewer_JLGy · 2024-11-03

**Soundness:** 3
**Presentation:** 3
**Contribution:** 3
**Rating:** 8
**Confidence:** 3

**Summary:**

This paper presents a novel pruning method named THINK to reduce KV cache memory size for long-context inputs. The authors observed a gap in optimizing the channel dimension for KV cache parameters. They show that THINK can reduce memory size by 20% with minimal accuracy loss.

**Strengths:**

- Timely problem
- Novelty
- Solid experimental results

**Weaknesses:**

- Comparison with related work

**Questions:**

Thank you for submitting to ICLR 2025. The research problem is both interesting and timely. The motivating examples are helpful in identifying existing research gaps in channel pruning. The solution is well-reasoned, resulting in significant memory improvements.

What are the authors' reasons for choosing the three baselines?

It would also be helpful to see the results of THINK combined with existing methods on the sequence length dimension and the layer dimension.

---

> ### Author Response · Authors · 2024-11-21
> **Response to Reviewer JLGy**
>
> We thank the reviewer for the comments and questions.
>
> >**Q1.** What are the authors' reasons for choosing the three baselines?
>
> **A1.** H2O and SnapKV are state-of-the-art KV eviction methods that optimize KV cache efficiency from the sequence length dimension, while KIVI is the state-of-the-art KV cache quantization method that focuses on optimization through precision reduction. In our work, we address KV cache optimization from the channel dimension. We selected H2O, SnapKV, and KIVI as baseline methods to demonstrate that our approach can integrate seamlessly with state-of-the-art KV cache compression techniques to further improve memory efficiency without degrading performance. By comparing against these methods, we highlight the complementary nature of our approach and its potential to enhance existing solutions.
>
> >**Q2.** It would also be helpful to see the results of THINK combined with existing methods on the sequence length dimension and the layer dimension.
>
> **A2.** Thank you for the suggestion. H2O and SnapKV are methods that optimize the KV cache along the sequence length dimension. We have already provided the results of our method combined with ThinK in Table 2 and Table 3 of the original manuscript. Regarding methods that optimize KV cache along the layer dimension, many such approaches [1][2] require retraining the models from scratch, which is not feasible during the rebuttal period. We acknowledge the importance of this direction and will explore it further in future work.
>
>  [1] Layer-Condensed KV Cache for Efficient Inference of Large Language Models
>
>  [2] Reducing Transformer Key-Value Cache Size with Cross-Layer Attention

---

### Official Review · Reviewer_cgqU · 2024-11-04

**Soundness:** 2
**Presentation:** 2
**Contribution:** 2
**Rating:** 8
**Confidence:** 4

**Summary:**

This paper presents ThinK, a query-dependent K cache pruning method for long-context LLM inference. ThinK takes advantage of the channel-wise sparsity property of K cache in existing KV cache tensors, and uses a greedy algorithm to remove insignificant channels and to optimize for the loss in attention weights after pruning. ThinK can be used together with many existing token eviction techniques (H2O) and KV cache compression techniques (KIVI). Experiment results show that on long-context benchmark tasks, ThinK achieves a comparable level of accuracy when used jointly with H2O or SnapKV, as compared to using H2O or SnapKV alone, but lower memory consumption.

**Strengths:**

`+` Channel-wise sparsity for K cache isn't a new thing, but this is (probably) one of the first papers that present a solid method for channel-wise K cache pruning.

`+` Compatibility with existing token eviction or cache compression methods.

`+` Solid and extensive evaluation. I am convinced by tables 2 and 3 that ThinK is good at detecting and pruning insignificant channels in the K cache, since the accuracy drop is very low even for a pruning ratio of 0.4 or 0.5, so this is a reasonably designed method for K cache compression.

**Weaknesses:**

`-` Unfortunately, V cache typically demonstrate token-wise sparsity, so this method cannot be generalized to V cache. To some people, this might seem like a smart engineering hack for a specific type of tensors with known distribution properties. **(Clear: The reviewer is clear with the presentation of section #2 after the rebuttal period)**

`-` The presentation of evaluation results and the choice of evaluation metrics are very concerning to me. There is an overwhelmingly large focus on accuracy numbers (Tables 2, 3, 4, 5, and 6), but one of the most important benefits enabled by KV cache compression/pruning is increased throughput and decreased TTFT --- These two metrics do not even appear in the evaluation. Note that reduced memory usage (Figure 3a) is only an intermediate benefit you care about: Eventually, the hope is that reduced memory usage can enable larger batch size, and consequently larger throughput. Though Figure 3b has some numbers on batch size, it does not show how ThinK has the potential of boosting inference throughput. **(Clear: The authors presented updated numbers on latency / throughput during the rebuttal period)**

`-` Minor issues: Typos on line 188 ("pruning matrix"->"pruning metrics"?); weird sentence on lines 214-215 ("Based on the observations ..."). **(Clear: The authors fixed these minor issues during the rebuttal period)**

**Questions:**

Please refer to "weaknesses".

**Details Of Ethics Concerns:**

This paper does not raise any ethics concerns.

---

> ### Author Response · Authors · 2024-11-21
> **Response to Reviewer cgqU**
>
> We thank the reviewer for the comments and questions.
>
> >**Q1.** V cache typically demonstrate token-wise sparsity, so this method cannot be generalized to V cache. To some people, this might seem like a smart engineering hack for a specific type of tensors with known distribution properties.
>
> **A1.** Thank you for the feedback. Based on our observations, the Value cache exhibits less sparsity than the Key cache in the channel dimension, which is consistent with the patterns we have analyzed. However, as shown in Table 8, we demonstrate that pruning 30% of both the Key cache and Value cache on LLaMA-3-8B-Instruct still results in acceptable performance. This indicates that the Value cache also has potential for pruning in the channel dimension, contrary to the assumption that it is strictly token-wise sparse.
>
> We respectfully disagree with the characterization of our method as a "smart engineering hack." Our approach is grounded in commonly observed patterns across various base models and benchmarks, demonstrating consistent effectiveness. The sparsity of Key cache channels exist across all layers of large language models on a variety of base models. Furthermore, our method does not rely on hyperparameter tuning (except for the pruning ratio), making it broadly applicable and not specifically designed for any particular tensor type. Additionally, our work is the first to identify and leverage redundancy in the channel dimensions of the KV cache, providing a promising direction for future research. This contribution highlights the versatility of our method and opens new avenues for advancements in efficient LLM inference. It is also worth noting that similar approaches in the literature leverage specific observations to optimize LLM efficiency. For instance, StreamingLLM [1] improves efficiency by addressing attention sinks in attention weights, while KIVI [2] targets outliers in Key cache channels. These examples demonstrate how targeted insights can lead to practical and impactful efficiency improvements, which is the foundational philosophy of our work.
>
> [1] Guangxuan Xiao, Yuandong Tian, Beidi Chen, Song Han, and Mike Lewis. Efficient streaming language models with attention sinks. arXiv preprint arXiv:2309.17453, 2023b.
>
> [2] Zirui Liu, Jiayi Yuan, Hongye Jin, Shaochen Zhong, Zhaozhuo Xu, Vladimir Braverman, Beidi Chen, and Xia Hu. Kivi: A tuning-free asymmetric 2bit quantization for kv cache. arXiv preprint arXiv:2402.02750, 2024b.
>
> >**Q2.** Though Figure 3b has some numbers on batch size, it does not show how ThinK has the potential of boosting inference throughput.
>
> **A2.** Thank you for this valuable suggestion. During the rebuttal period, we conducted additional evaluations and now provide comparisons of TTFT (Time To First Token), TPOT (Time Per Output Token), memory usage, and throughput in the following tables. As our method performs online pruning, there is a slight delay introduced during the prefilling stage due to the computation of channel importance. However, the reduction in memory usage leads to notable improvements in TPOT during inference. Specifically, after applying ThinK, TPOT improves from 0.27 ms/token to 0.25 ms/token with 40% pruning, and further improves to 0.24 ms/token with a 50% pruning ratio. Regarding throughput, with the same memory usage, ThinK (40% pruning) improves the throughput of KIVI from 5168 tokens/s to 5518 tokens/s. At a pruning ratio of 50%, throughput increases further to 5676 tokens/s. These results highlight the potential of ThinK to enhance inference performance by balancing memory efficiency with computational overhead. We are continuing to optimize the implementation to further improve performance and will include these new results and analyses in the revised manuscript.
>
> |**Methods**|**KIVI(4/4)**|**KIVI(4/4)+ThinK(0.4)**|**KIVI(4/4)+ThinK(0.5)**|
> |--------------|--------------|--------------|--------------|
> |**TTFT(ms)**| 7 | 10.4|10.2|
> |**TPOT(ms/token)**|0.27|0.25|0.24
> |**Mem(GB)**|61.7|53.3|51.2|
>
> Table. Comparisons of TTFT (Time To First Token) and TPOT (Time Per Output Token).
>
> |**Methods**|**Vanilla**|**KIVI(4/4)**|**KIVI(4/4)+ThinK(0.4)**|**KIVI(4/4)+ThinK(0.5)**|
> |--------------|----------------------|--------------|--------------|--------------|
> |**Throughput(token/s)**|2557|5168|5518|5676|
>
> Table. Throughput comparisons of different methods.
>
> >**Q3.** Minor issues: Typos on line 188 ("pruning matrix"->"pruning metrics"?); weird sentence on lines 214-215 ("Based on the observations ...").
>
> **A3.** Thanks, we have updated in the revision.

---

> ### Author Response · Authors · 2024-11-25
> **Looking forward to your reply**
>
> Dear Reviewer cgqU,
>
> We hope this message finds you well. As the author-reviewer discussion period approaches, we respectfully seek your confirmation on the adequacy of our rebuttal in addressing the concerns raised in your review.
>
> We really appreciate the substantial time and effort you have committed to reviewing our work and are grateful for the additional insights. Your comments have been very helpful in refining our project.
>
> Thank you once again for your valuable perspectives. We eagerly await your further guidance.
>
> Sincerely,

---

> ### Comment · Reviewer_cgqU · 2024-11-27
>
> Hi,
>
> Thanks for your reply as well as your efforts in preparing the updated version of the paper. I gave the updated version and all other reviews a closer read, and here are some thoughts:
>
> Regarding weakness #1, I agree (from the beginning) with the authors' comment that the designed approach is motivated by empirical observations regarding the K cache tensor, rather than a random trick (apparently I missed figure 4, etc.) --- I hope that you did not understand it as if I was de-valuing your work. On this front, I think the paper (section #2) is presenting the analysis in a scientific way, which I appreciate.
>
> It seems to me that the remaining concern is the tremendous amount of concurrent work in this space that might be presenting similar empirical observations (that K cache is showing channel-wise sparsity). At this point, I don't think that should be used as a reason for turning down a paper --- However, if this paper eventually got in, it would be incredibly helpful if the authors could clarify how the empirical observations with the K cache tensor differ from (or overlap with) concurrent work (like those mentioned by ssq7).
>
> Weakness #2: Clear. The throughput and latency numbers are incredibly helpful. My understanding is that: these system performance numbers are what most users would pay attention to if they were to select a certain KV cache compaction method.
>
> The ideal case, in fact, is to present visualizations of accuracy-performance trade-offs, e.g. in the form of dot/line plots. That would show an even more direct comparison of ThinK v.s. existing methods (w/o using ThinK), e.g. ThinK might give a better overall trade-off, consuming much less memory but yielding almost the same level of accuracy. I don't think these visualization results are strictly necessary at this point, but would be nice to have if this paper got in.
>
> At this point, I don't have particular concerns with this paper, and I have raised my score.
>
> Reviewer cgqU

---

> > ### Author Response · Authors · 2024-11-27
> >
> > Thank you for acknowledging our work and for your thoughtful feedback.
> >
> > We appreciate your suggestion to clarify how the empirical observations regarding the Key cache tensor differ from or overlap with concurrent work. We will include this discussion in the final version of the paper.
> >
> > Thank you once again for your suggestions and support.

---

### Official Review · Reviewer_ELRT · 2024-11-04

**Soundness:** 3
**Presentation:** 3
**Contribution:** 2
**Rating:** 5
**Confidence:** 4

**Summary:**

The paper presents a KV cache optimization method with channel wise pruning for reducing the inference cost under long context scenario. The method is based on the previously proposed efficient context compression method ( H2O, KIVI, and SNAP). Specifically, the paper utilized a query driven method to involve the query for key channel selection.

**Strengths:**

The proposed method is simple and straight forward, with direct structured pruning on the key channel.
The evaluation shows the pruned results achieves comparable results with the non-pruned baselines

**Weaknesses:**

Lack of experiments compared to directly applying to vanilla models.
Why the proposed method is basing on the pruned method, instead of directly on the vanilla model? The observations come from the fact of that there are large outliers alongside the channel. This not an unique attribute on the compressed model (i.e. H2O). Yet, experimenting on the compressed model introduces confounding variables which affect the analysis. The current manuscript lacks experiments directly apply the method on the Vanilla model.

In the meantime, the current manuscript lacks the details about how the K cache selected K cache is removed. This is important in order to consider the real memory benefits. Is the K cache fully removed? Or full of the KV cache are always stored in memory? If the selected channel of the K cache is removed, when new queries come, how to upgrade to the new selected K cache size? If the whole K cache are always stored in memory, how is the memory efficiency comes from?

It is also worth to indicating whether the compression is done starting at the prefilling phase, or the prefill phase is concrete. The experiments of many the previous approach (i.e. H2O) did not compress during prefill phase. Detailing it will make the understanding of the experimental results more comprehensive.

Typos:
Sec 3.1 line 149 KV cach
Line 175 k is bold?

**Questions:**

See weakness section.

---

> ### Author Response · Authors · 2024-11-21
> **Response to Reviewer ELRT (1/2)**
>
> We thank the reviewer for the comments and questions.
>
> >**Q1.** Lack of experiments compared to directly applying to vanilla models.
>
> **A1.** Thank you for your insightful suggestion. In response, we have conducted additional experiments applying ThinK directly to vanilla models, specifically LLaMA-3-8B-Instruct and Mistral-7B-Instruct-v0.2. The results, summarized in the following tables, demonstrate that our method maintains superior performance even after pruning 40% of the key cache channels. Furthermore, when the pruning ratio is increased to 50%, the performance degradation remains within an acceptable range. These findings further validate the effectiveness of ThinK, not only for pruned models but also when applied directly to vanilla models. We hope this addresses your concern. **In the revised paper, we have added the following tables.**
>
> | **Method**|**$\lambda$**|**Avg.** |**NrtvQA** | **Qasper** | **MF-en** | **HotpotQA** | **2WikiMQA** | **Musique** | **GovReport** | **QMSum** | **MultiNews** | **TREC** | **TriviaQA** | **SAMSum** | **PCount** | **PRe** | **Lcc** | **RB-P** |
> |--------------|----|-----|-----|------------|-----------------|---------------|-------------|---------------|-----------|---------------|----------|--------------|-------------|------------|---------|----------|-----------|-------|
> |LLaMA3 |-|41.86|25.56 | 32.27 | 39.71 | 43.56 | 35.09 | 21.18 | 28.71 | 23.26 | 26.64 | 73.50 | 90.48 | 42.33 | 4.80 | 69.25 | 59.29 | 54.05 |
> |LLaMA3+THINK|0.4|42.27|25.32 | 32.26 | 39.81 | 44.19 | 34.77 | 21.10 | 28.63 | 23.13 | 26.38 | 73.50 | 90.58 | 41.69 | 5.21 | 69.50 | 61.94 | 58.37 |
> |LLaMA3+THINK|0.5|42.25|25.35 | 32.80 | 41.64 | 43.99 | 31.81 | 21.58 | 28.35 | 23.31 | 26.80 | 73.50 | 90.37 | 40.81 | 5.67 | 69.17 | 61.90 | 59.00 |
> |LLaMA3+THINK|0.6|41.43|24.39 | 31.16 | 41.80 | 42.52 | 31.63 | 20.70 | 25.84 | 23.18 | 25.46 | 73.50 | 90.43 | 38.97 | 5.77 | 68.46 | 59.63 | 59.38 |
>
> Table. Performance comparison of Key cache pruning on LLaMA-3-8B-Instruct on LongBench.
>
> | **Method**|**$\lambda$**|**Avg.** |**NrtvQA** | **Qasper** | **MF-en** | **HotpotQA** | **2WikiMQA** | **Musique** | **GovReport** | **QMSum** | **MultiNews** | **TREC** | **TriviaQA** | **SAMSum** | **PCount** | **PRe** | **Lcc** | **RB-P** |
> |--------------|----|-----|-----|------------|-----------------|---------------|-------------|---------------|-----------|---------------|----------|--------------|-------------|------------|---------|----------|-----------|-------|
> |Mistral |-|42.71|26.63 | 32.99 | 49.34 | 42.77 | 27.35 | 18.77 | 32.87 | 24.24 | 27.10 | 71.00 | 86.23 | 42.96 | 2.75 | 86.98 | 56.93 | 54.49 |
> |Mistral+THINK|0.4|42.76|27.11 | 33.46 | 48.73 | 41.79 | 28.14 | 18.87 | 32.45 | 24.55 | 27.09 | 71.00 | 86.26 | 43.02 | 3.95 | 86.31 | 56.99 | 54.36 |
> |Mistral+THINK|0.5|42.53|26.63 | 33.71 | 49.38 | 42.38 | 26.78 | 18.76 | 32.57 | 24.63 | 26.92 | 71.00 | 86.39 | 42.82 | 3.13 | 84.65 | 56.75 | 54.04 |
> |Mistral+THINK|0.6|42.16|27.03 | 33.23 | 49.49 | 42.65 | 26.43 | 18.14 | 31.74 | 24.75 | 26.57 | 71.00 | 86.28 | 41.40 | 3.50 | 83.11 | 55.93 | 53.37 |
>
> Table. Performance comparison of Key cache pruning on Mistral-7B-Instruct-v0.2 on LongBench.

---

> ### Author Response · Authors · 2024-11-21
> **Response to Reviewer ELRT (2/2)**
>
> >**Q2.** Lack the details about how the K cache selected K cache is removed. This is important in order to consider the real memory benefits. Is the K cache fully removed? Or full of the KV cache are always stored in memory? If the selected channel of the K cache is removed, when new queries come, how to upgrade to the new selected K cache size? If the whole K cache are always stored in memory, how is the memory efficiency comes from?
>
> **A2.** Thank you for pointing out the need for more clarity regarding the handling of the Key cache. Detailed implementation is provided in Section 3.3, but we would like to elaborate further here. Our method removes unimportant Key cache channels. This pruning is motivated by the one-to-one correspondence between the Key and Query channels in the attention mechanism, as shown in the following formula: $\text{Attention}(\textbf{Q},\textbf{K},\textbf{V}) = \text{softmax}(\frac{\textbf{Q}\textbf{K}^T}{\sqrt{D}})\textbf{V}$. When a Key cache channel is removed, its corresponding Query channel is also removed, ensuring consistent dimensionality between the two. This query-driven pruning mechanism is the foundation of our approach.
>
> To address memory efficiency: the removed Key cache channels are not stored in memory, leading to a significant reduction in memory usage. Instead, we store a binary mask as an indicator of the pruned channels, and its memory cost is negligible. When new queries arrive, we dynamically apply the stored mask to prune the Query channels, ensuring that the new Query dimensionality and the pruned Key cache remain consistent. Since previously removed channels are not reintroduced, the memory savings are both significant and consistent over time. **We have updated Section 3.3 for greater clarity. Additionally, we have provided the implementation code of our method in the supplementary materials for transparency and reproducibility.**
>
> >**Q3.** It is also worth to indicating whether the compression is done starting at the prefilling phase, or the prefill phase is concrete. The experiments of many the previous approach (i.e. H2O) did not compress during prefill phase. Detailing it will make the understanding of the experimental results more comprehensive.
>
> **A3.** Thank you for highlighting this important point. We apologize for any confusion caused. Similar to SnapKV, our method compresses the Key cache starting from the prefilling stage to address the memory challenges associated with long-context inputs. We include this clarification in the implementation details of the revised manuscript to ensure the experimental setup and results are more comprehensively understood.
>
> >**Q4.** Typos: Sec 3.1 line 149 KV cach Line 175 k is bold?
>
> **A4.** Thanks, we have updated in the revision

---

> ### Author Response · Authors · 2024-11-25
> **Looking forward to your reply**
>
> Dear Reviewer ELRT,
>
> We hope this message finds you well. As the author-reviewer discussion period approaches, we respectfully seek your confirmation on the adequacy of our rebuttal in addressing the concerns raised in your review.
>
> We really appreciate the substantial time and effort you have committed to reviewing our work and are grateful for the additional insights. Your comments have been very helpful in refining our project.
>
> Thank you once again for your valuable perspectives. We eagerly await your further guidance.
>
> Sincerely,

---

### Author Response · Authors · 2024-11-21
**Paper Update Summary**

We thank the area chair and the reviewers for their valuable comments.
We also revised the paper and updated the PDF file in the OpenReview system. In our current version, the revisions in the main article are marked in magenta. We summarize the revised paper details.

 1. Experiment result of ThinK on vanilla models are added in Table 9. (**Reviewer ELRT**)
 2. Fix typos on line 149 and line 176. (**Reviewer ELRT**)
 3. Clarify the implementations in Section 3.3. (**Reviewer ELRT**)
 4. TTFT, TPOT, and Throughput results are added in Table 10 and Table 11. Discussions are added in Appendix F. (**Reviewer cgqU**)
 5. Fix typos on line 188 and on lines 214-215. (**Reviewer cgqU**)
 6. Comparisons of ThinK and SVD based methods are added in Table 12. Discussions are included in Appendix G. (**Reviewer JLGy**)
 7. Explanation of pruning ratio are added in Section 4.2 on page 7. (**Reviewer JLGy**)
 8. Discussions about value cache are added in Section 4.4 on page 10 and Appendix D. (**Reviewer JLGy**)
 9. Add notation in Section 4 on Page 6. (**Reviewer 1rme**)
 10. Clarify the contributions in Abstract, Section 4.4 on page 9 and Section 6. (**Reviewer 1rme**)
 11. Add discussions on how can benefit inference in Section 4.4 on page 9 and Appendix F. (**Reviewer 1rme**)
 12. Add anonymized codes as the supplementary file to enhance reproducibility.

Apart from the issues addressed above, we also revised our paper with regard to consistency and readability.

---

### Meta-Review · Area_Chair_XL5J · 2024-12-21

**Metareview:**

The paper introduces a novel query-dependent KV cache pruning method to reduce memory usage during LLM inference. By targeting channel sparsity in the Key cache, ThinK achieves over 20% memory reduction and 2.8× peak memory savings with minimal accuracy loss, validated across extensive benchmarks. Strengths include the method’s simplicity, compatibility with existing optimizations, and robust experimental results. Weaknesses include limited exploration of Value cache pruning and insufficient baseline comparisons. Despite these, the paper presents a significant contribution to memory-efficient LLM deployment, making it a valuable addition to the field.

**Additional Comments On Reviewer Discussion:**

The author-reviewer discussions highlighted a balanced set of strengths and areas for improvement. The authors effectively addressed reviewer concerns by providing additional experimental results, clarifying the handling of Value cache pruning, and presenting detailed evaluations on latency and throughput. These responses strengthened the paper by resolving ambiguities and improving clarity. Reviewers agreed on the paper's significance, noting that the limitations, such as dependency on Key cache pruning and increased TTFT, are acceptable given the practical advantages. It is recommended that the authors explicitly outline these limitations in the final version and explore further integration with Value cache pruning to maximize the method's impact.

---

### Decision · Program_Chairs · 2025-01-22

Accept (Spotlight)